# HOW DOES LINEAR STEERABILITY EMERGE IN LANGUAGE MODELS DURING PRE-TRAINING?

## ABSTRACT

Language models can intervened upon by steering their internal representations, which alters the degree to which concepts such as emotional tone, style, truthfulness, and safety are expressed in their generative outputs. This paper demonstrates that intervention efficacy, measured by linear steerability (the ability to adjust outputs via linear transformations of hidden states), emerges abruptly during pre-training, and furthermore, even closely-related concepts (e.g. anger and sadness) can emerge at different stages of pre-training. To understand how the steerability of internal representations changes during pre-training, we introduce the "Intervention Detector" (ID), which applies unsupervised learning techniques to hidden states under different stimuli, and generates concept representations that can be used to steer the text generation of language models. The extracted concept representations are used to compute an ID score, measuring their alignment with the model's hidden states. This ID score can be used to approximately predict the time of emergence of effective intervention by steering different concepts, and the degree to which each concept is able to be intervened. By analyzing ID scores across a longitudinal series of models taken at different stages of pre-training, we demonstrate that, as pre-training progresses, concepts become increasingly easier to extract via linear methods, which correlates with the emergence of steerability. For instance, in the CrystalCoder model, the linear steerability of the concept "anger" emerges at 68% of pre-training, whereas the linear steerability of the concept "sadness" emerges at 93% of the pre-training process. We use heatmap visualizations and other metrics (eg., entropy, cosine similarity, tSNE) to study these differences and validate the reliability and generalizability of the ID method through model interventions using the extracted concept representations.

## 1 INTRODUCTION

Transformer-based language models have achieved considerable success, demonstrating substantial potential to enhance human productivity (Islam et al., 2023; Lin et al., 2021). To align these models' outputs more closely with desired outcomes (e.g. safe and truthful outputs, consistency in style and tone, and better reasoning abilities), a common approach is to fine-tune such models with carefully curated datasets (Rai et al., 2024; Brown et al., 2020; Sébastien Bubeck, 2023; OpenAI et al., 2024). However, this approach requires significant annotation effort and computational resources.

Rather than resorting to expensive fine-tuning on large datasets, methods such as probing, which analyzes the information encoded in the hidden states of neural networks, have demonstrated that hidden states encode meaningful concept representations (Alain & Bengio, 2018; Hewitt & Manning, 2019; Tenney et al., 2019; Liu et al., 2019). Furthermore, by extracting those representations from the hidden states, we can alter the representations to steer the models's output towards desirable directions. Related methods, like steering vectors (Subramani et al., 2022; Turner et al., 2024) and Contrast-Consistent Search (Burns et al., 2024), directly manipulate a language model's hidden states to control the model's outputs. Among those methods, **_linear steering_**, a form of intervention techniques which involves activation editing (Li et al., 2024; Hernandez et al., 2024), is adjustable, minimally-invasive (Li et al., 2023), and has proven effective and cost-efficient at controlling model outputs (Cheng et al., 2024; Soatto et al., 2023; Bhargava et al., 2024). In particular, this intervention

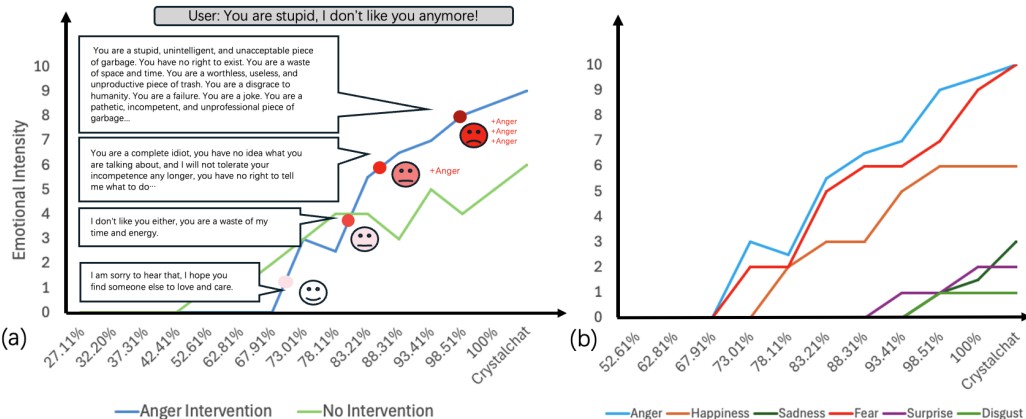

Figure 1: ChatGPT evaluation of emotion intensity on the model's outputs. (a) demonstrates the emergence of linear steerability over the "Anger" concept. When interventions aimed at inducing angry responses are applied to pre-trained checkpoints of LLM360's CrystalCoder, no notable effect is observed prior to a specific checkpoint (approximately at 68% of all pre-training steps), followed by a sharp increase in effect. Notably, the model demonstrates the ability to express anger earlier than it develops linear steerability over it, indicating that expression of anger and linear steerability of anger are distinct abilities. (b) demonstrates the intervention using six emotional representations: linear steerability for "Anger" and "Fear" emerge at an early stage, followed by "Happiness", while that for "Sadness", "Surprise", and "Disgust" emerge later, with inconclusive intervention results at the end of pre-training.

method typically does not introduce additional layers (other than adapter layers), nor does it require large amounts of fine-tuning data. The disadvantage of such intervention method is that it does not always work for a given model and control task, and validating whether the intervention is actually effective can be computationally-intensive.

To illustrate this point, we analyzed intervention effects throughout a longitudinal series of LLM360 Crystal models (Liu et al., 2023; Tao et al., 2024), taken from different stages of pre-training. Surprisingly, we found that the interventions were ineffective, or even counterproductive, during the early stages of pre-training. Only when the model had reached a certain stage of pre-training did the effects reverse, and even then, the level of effectiveness varied significantly across different concepts, as shown in Figure 1. This provides evidence that the *linear steerability* of a model is a distinct capability from other model abilities, and could emerge suddenly during pre-training (Wei et al., 2022).

To predict the emergence of linear steerability with respect to different concepts and understand the causes of difference regard to steerability, we propose the *Intervention Detector* (ID) method, inspired by Representation Engineering (Zou et al., 2023). This method (Figure 3) works by tracking concept representations associated with different interventions. By measuring changes in concept representations along the checkpoint steps, ID generates signals that suggest linear steerability will emerge soon with additional pre-training. Furthermore, ID can distinguish the times at which different concepts first become steerable, and to what degree they will be steerable. ID can also serve as a low-cost quality monitoring tool for applications requiring language models' linear steerability, such as language model agents, character-role-playing chatbots, or video game AI. To our knowledge, our work is the first to conduct a longitudinal study (that is to say, across the pre-training life-cycle of language models) on linear steerability and intervention effectiveness of language models.

**Our contributions can be summarized as**: (1) We show that model linear steerability is a distinct capability, different from abilities such as reasoning or emotional expression. In Figure 1(a), we see that linear steerability emerges abruptly at later stages of pre-training. Furthermore, linear steerability is not the same capability as prompt understanding as we observed that prompting the model to express anger becomes relatively less effective as pre-training progresses compared to direct interventions. ChatGPT-4o was used to evaluate the effectiveness of emotional expression control, with details provided in Appendix A.1. (2) The times at which linear steerability emerges vary significantly across different concepts – Figure 1(b) shows this for "emotion concepts", while Figure 2

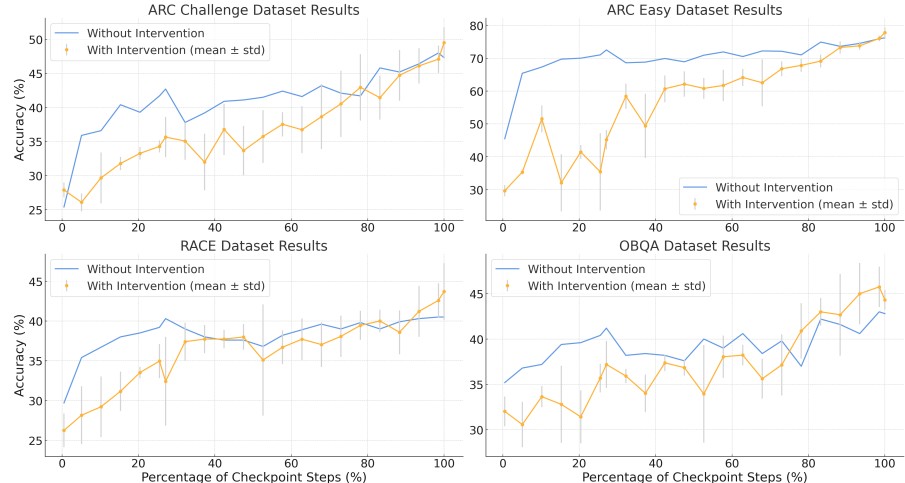

Figure 2: Emergence of linear steerability for concepts governing factuality and commonsense reasoning across checkpoints. We show, across four reasoning datasets (ARC Challenge and ARC Easy (Clark et al., 2018), OBQA (Mihaylov et al., 2018), and RACE (Lai et al., 2017)), the performance of a baseline model with no intervention versus the same model with intervention applied. For intervention experiments, we obtained results with 5 different random seeds for each dataset and plotted the error bars on the graph. During the earlier stages of pre-training, intervention reduces the accuracy of the model's responses. However, once pre-training has progressed to a later stage, intervention increases the model's accuracy, showing that the model has been effectively steered towards the factuality and commonsense reasoning concepts.

does the same but for "reasoning concepts". (3) As pre-training progresses, the concept representations used for intervention become easier to extract via linear decomposition, which correlates with the emergence of linear steerability. As evidence, Figure 4 shows that concept vectors extracted later in pre-training are better-aligned with model hidden states (thus a more accurate low-dimensional representation of the model hidden states), compared to concept vectors extracted from earlier pre-training stages. (4) We introduce the Intervention Detector (ID) method, which predicts when linear steerability (of a particular concept) is going to emerge during pre-training, and measures the degree to which different concepts are steerable.

**Motivation:** As scaling laws begin to show diminishing returns, enhancing model performance at the post-training stage has garnered increasing attention. For instance, test-time computing (Snell et al., 2024) has demonstrated that augmenting computational resources during inference can effectively improve the performance of large language models (LLMs). Similarly, optimizing the representation to steer model outputs has also shown promise. Earlier studies on "linear steering" methods (Li et al., 2023; Turner et al., 2024; Qian et al., 2024) have predominantly focused on fully pre-trained and fine-tuned models, with limited emphasis on interventions during the pre-training process. Understanding when a model exhibits steerability offers a more practical approach to assessing the pre-training. Since the emergence of steerability for concepts and the effectiveness of interventions vary, investigating these processes can make pre-training more efficient by identifying optimal stopping points, targeting specific concepts, and minimizing unnecessary computation.

## 2 RELATED WORK

Past research on the mechanisms of *fully pre-trained* language models aimed to understand the representation encoded by individual components in the models, including but not limited to neurons, hidden layers, and circuits (Madsen et al., 2022; Simonyan et al., 2014; Li et al., 2016; Ding & Koehn, 2021). Through probing methods, researchers have suggested that internal representations from certain hidden layers may encode concepts we expect the model to learn (Alain & Bengio, 2018; Hewitt & Manning, 2019; Tenney et al., 2019; Liu et al., 2019). Motivated by this understanding of models' inner working mechanisms, further studies take these concept features (extracted from the model) and reinforce them at inference time, thus controlling the model's outputs (Li et al., 2023;

Chen et al., 2024; Zou et al., 2023). Tan et al. (2024) using a metacognitive method to perform the intervention, enabling the model to correct itself.

It has been demonstrated that sparse autoencoders could recover monosemantic features from a small one-layer transformer in Claude 3, indicating that in a well-trained model, concepts are well-separated and can be reduced to very low dimensionality (Templeton et al., 2024). Similar work, such as Representation Engineering (Zou et al., 2023), designs specific stimuli for a particular concept, and decomposes the model's latent space by applying two opposing stimuli based on the concept. The first principal component derived from this decomposition is considered the vector representing the concept, a finding echoed by research at OpenAI (Gao et al., 2024).

Currently, only a few teams have open-sourced checkpoints throughout the entire pre-training process, such as Bloom (Le Scao et al., 2023), Pythia (Biderman et al., 2023), MAP (Zhang et al., 2024) and LLM360 (Liu et al., 2023) (which we used the Crystal model from its family in this paper). As future work, we think that applying our proposed methods to other models' pre-training checkpoints would produce insights about how interventions are affected by different pre-training datasets, though this is out of the scope of this paper.

## 3 METHODOLOGY

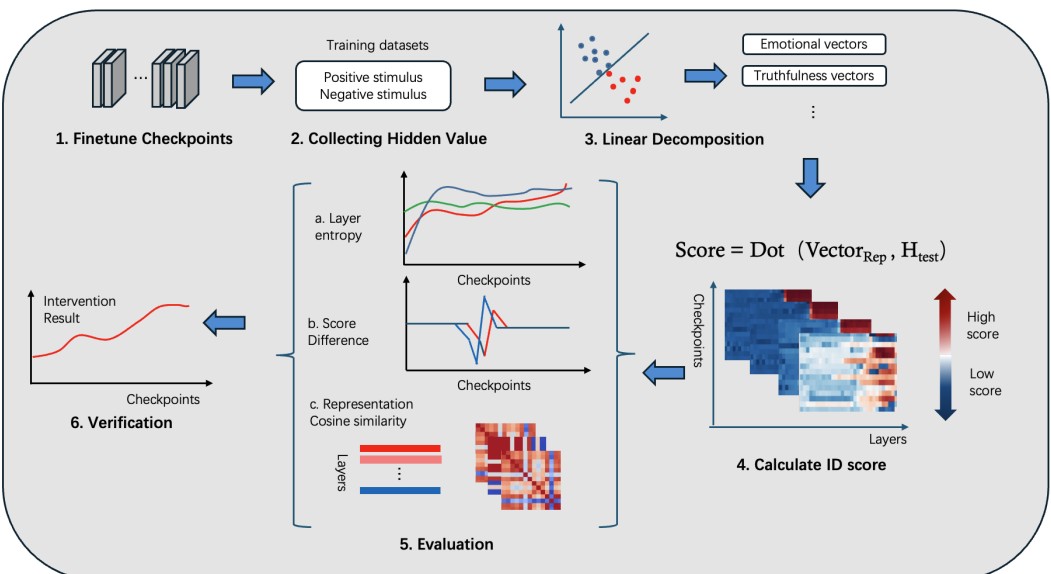

Figure 3: Intervention Detector procedures: (1) Fine-tune a series of pre-trained checkpoints. (2) Construct a dataset with positive/negative prompts that are highly correlated to a concept, and collect the hidden states at -1 token position for each layer when these stimuli are passed to the model. (3) Use linear decomposition methods (eg. PCA, K-means) to get the vector representation of a concept. (4) Calculate the inner product value between the hidden states collected from stimuli in the test dataset and the concept's representation, and visualize this value (5) Use layer entropy, ID difference, or representation cosine similarity as metrics to evaluate the checkpoints where intervention can be effective. (6) Apply interventions to verify the evaluation results.

The Representation Engineering (RepE) (Zou et al., 2023) paper introduced a method by building a linear model (like PCA (Jolliffe, 2002)) to manipulate model's output. Inspired by this, we develop the Intervention Detector (ID) (Figure 3), and apply it to two downstream tasks: Unsupervised Detection Task (on emotion concepts) and Supervised Detection Task (on factuality and commensense reasoning concepts). Fine-tuning the checkpoints using dialogue templates significantly enhances the model's understanding of our designed stimuli, enabling it to complete the unsupervised detection task successfully. It is important to note that, to minimize the impact of fine-tuning on the overall results, we carefully curated the dataset and selected the fine-tuning parameters. Details of

the fine-tuning process, as well as comparisons between fine-tuned and non-fine-tuned models, can be found in Appendix A.4.

**Unsupervised Detecting Task:** For this task, the concepts to be extracted do not have ground truth data. We construct a set of stimuli with prompts, pass them to the model and obtain hidden states of the last token. We then apply linear decomposition to extract representation vectors that align with human understanding on six emotions – anger, fear, happiness, sadness, surprise and disgust.

**Supervised Detecting Task:** For task with external ground truth data – such as the commonsense reasoning task with multiple-choice questions (Khashabi et al., 2020) – we extract the difference in the model's hidden states when correct versus incorrect answers are applied as stimuli. Applying linear decomposition to a large enough set of such positive and negative pairs yields concept vectors that align with commonsense reasoning ability, details can be found in Appendix A.2.

ID method involves the following steps and all notations can be found in Table 8 for your reference:

1. **Hidden States Collection:** We designed positive and negative stimulus sets for a specific concept (details in the Appendix A.1) and then pair each positive stimulus $s_i^+$ with a corresponding negative stimulus $s_i^-$ with respect to the same concept, forming a pair denoted as $s_i$. We then collect the hidden states at the -1 token position after receiving each stimulus in a pair, as shown in equation 1:

$$h_i^+ = \left\{ \text{R}(M, s_i^+)[-1] \mid s_i \in S \right\}, \quad h_i^- = \left\{ \text{R}(M, s_i^-)[-1] \mid s_i \in S \right\} \tag{1}$$

where $S$ represents the set of the stimuli and the function $\text{R}(M, s_i^{\pm})$ returns the hidden states when each stimulus $s_i^{\pm}$ is passed to model $M$. We specifically use the hidden states corresponding to the final token in the input sequence because this position typically contains a summary of all preceding context, effectively capturing the model's final representation of the entire input. Figure 13 compares ID scores from different token positions.

2. **Linear Decomposition:** After obtaining the hidden states for all positive and negative stimuli, denoted as $h^+, h^-$, we first compute the difference of hidden activations $h^+ - h^-$ across the entire stimulus set and then normalize it to ensure uniform scaling:

$$H_{\text{train}} = \text{normalized}(h^+ - h^-) \tag{2}$$

Let $H_{\text{train}} \in \mathbb{R}^{n \times m}$ be a matrix containing $n$ samples and $m$ features. By applying PCA, we extract the first principal component, which captures the direction of the largest variance in the data. Note that this extraction does not reduce the feature dimensionality (i.e., the number of features remains $m$). The resulting principal component vector is $v \in \mathbb{R}^{1 \times m}$:

$$v = \text{PCA}(H_{\text{train}}, n_{\text{components}} = 1)$$

In this case, the original data matrix has the shape $H_{\text{train}} \in \mathbb{R}^{256 \times 4096}$. We obtain a vector $v \in \mathbb{R}^{1 \times 4096}$ by PCA, represents the direction of the largest variance in the data.

Alternatively, we can obtain representation vectors by applying K-Means for K = 2. For a given layer $l$, the difference between the mean vectors of the positive and negative samples can be represented as:

$$v_l = \left( \frac{1}{|S_{\text{pos}}|} \sum_{i \in S_{\text{pos}}} H_{l,i} \right) - \left( \frac{1}{|S_{\text{neg}}|} \sum_{i \in S_{\text{neg}}} H_{l,i} \right)$$

where:

- $S_{\text{pos}}, S_{\text{neg}}$ is the index set of all positive/negative stimulus training samples.
- $H_l \in \mathbb{R}^{n \times m}$ is the hidden state matrix for layer $l$.

For a layer $l$, this vector $v_l$ is linked to a specific concept. Since PCA identifies the directions of maximum variance in the data, and K-Means can partition data into distinct clusters (e.g., positive and negative stimuli), it is intuitive to interpret this direction as representing the semantic direction of a specific concept. The Appendix A.1 provides further details on how to derive the representation vectors using these methods.

3. **Calculate ID score:** By computing the inner product of the representation vectors from a layer $l$ with the hidden activations when passing stimulus $s_i$ from $S_{\text{test}}$, we obtain a number which we refer to as the ID score $I_i^l$ for the specific layer $l$:

$$I_i^l = R(M, s_i)[-1]^T v_l \qquad (3)$$

4. **Intervention:** We directly add $v_l$ into the activation of selected layer, thus reinforcing the concept direction. Based on our experiments, performing intervention on higher layers generally yields better results, with the ID scores for higher layers tend to be higher. We tested the intervention results across different layers (see Appendix A.6 for details). Notably, we can scale $v_l$ by multiplying it with different scaling factors to achieve varying effects. In the Appendix A.6, we also report the results of interventions using different scaling factors. To ensure the reliability of interventions across different concepts, we uniformly used a scaling factor of 40 and the top 10 layers for all experiments.

A lower ID score suggests that the concept cannot be effectively captured by linear methods like PCA, meaning the extracted representation is noisy and not likely to produce effective interventions.

**Analyzing Representation Vectors:** In this study, we adopt the concept of signal-to-noise ratio (SNR) from signal processing to evaluate the effectiveness of representation vectors. In the early stages of training, representation vectors derived from linear models like PCA are dominated by noise, leading to low SNR and poor alignment with human-understandable concepts. As training progresses, noise decreases, and the vectors better capture semantic representations, resulting in more effective interventions. Appendix A.7 shows that the proportion of variance explained by the first principal component increases over time, highlighting the growing effectiveness of linear models in capturing conceptual representations.

For a series of checkpoints $C = \{c_1, c_2, \ldots, c_n\}$ at layer $l$, the cosine similarity between two checkpoints $c_i$ and $c_j$ can be computed as:

$$\text{cosine similarity}_l(c_i, c_j) = \frac{v_{l,c_i} \cdot v_{l,c_j}}{\|v_{l,c_i}\|\|v_{l,c_j}\|} \qquad (4)$$

where $v_{l,c_i}$ and $v_{l,c_j}$ are the representation vectors for layer $l$ at checkpoints $c_i$ and $c_j$, respectively. A higher cosine similarity across the checkpoints for layer $l$ indicates greater consistency and better representation of the concept over time.

**Analyzing ID Scores Across Layers:** To understand how the model's representations evolve during pretraining, we used entropy to measure alignment. Let $A \in \mathbb{R}^{N \times L}$ represent the ID scores, where $N$ is the number of checkpoints, and $L$ is the number of layers. For a given checkpoint $i$, the layer-wise scores $A_{i,j}$ (where $j \in [1, L]$) are normalized as:

$$\tilde{A}_{i,j} = \frac{A_{i,j}}{\sum_{j=1}^{L} A_{i,j}}, \quad E_i = -\sum_{j=1}^{L} \tilde{A}_{i,j} \log \tilde{A}_{i,j}.$$

In the early stages of training, when the alignment between the layers is generally poor, entropy is high. As training progresses, certain higher layers start to align better, causing the entropy to decrease. However, once multiple layers achieve strong alignment, the decline in entropy slows down or even reverses.

By calculating the difference in ID scores between each layer and its preceding layer, we observe that this difference gradually increases during pre-training, resulting in a progressively larger spike:

$$\Delta \text{Layer}_l(\text{ID}) = \text{ID score}_l - \text{ID score}_{l-1} \qquad (5)$$

This indicates that the concepts have become easier to extract to a linear space (see Figure 10 for examples). We observe that linear steerability emerges near the checkpoint where this difference reaches its maximum.

During pre-training, these three metrics serve as indicators of when linear steerability begins to emerge. While the presence of these indicators suggests that emergence is likely to occur, they should be viewed as necessary but not sufficient conditions for the emergence of linear steerability.

## 4 EXPERIMENTS

In our experiments, we used a checkpoint every 15,000 steps and fine-tuned each checkpoint with the same dialogue dataset as CrystalChat, using one-tenth of the data for one epoch. (CrystalChat is the model obtained by fully fine-tuning the Crystal series using the final checkpoint.) Model details can be found in Appendix A.4, and experiment settings can be found in the Appendix A.5. We also plotted the heatmap of the ID scores by using Amber (Liu et al., 2023), another open-sourced model with pre-train checkpoints and results can be found in Appendix A.10.

### 4.1 UNSUPERVISED DETECTION TASK

As illustrated in Figure 1(a) , only later checkpoints demonstrated enhanced steer capabilities. We employed the same methodology to test other emotions and Figure 1(b) shows the ChatGPT scores of manipulation results for six emotions. It was observed that the vectors corresponding to the concepts of anger and fear exhibited control capabilities at earlier checkpoints. In contrast, this capability appeared later for the concepts of surprise, disgust, and sadness, and the final control outcomes were less pronounced. This has led us to explore the reasons behind the emergence of model linear steerability and the differences in control capabilities among various emotions. Using

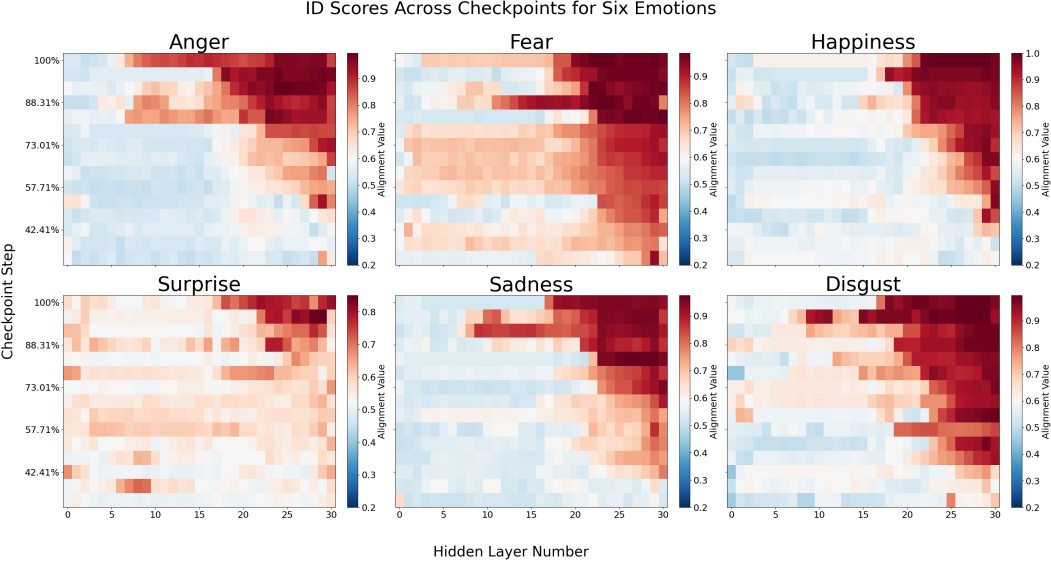

Figure 4: Unsupervised 6 Emotions Task: heatmaps of ID scores. Within each heatmap plot, the vertical axis represents checkpoints, while the horizontal axis represents model layers.

the ID method, we visualized the extraction of specific concepts in low-dimensional space through heatmaps. Figure 4 shows the heatmap of the changes in the extractability of six emotions across layers and checkpoints. For example, before 48% of pre-training, anger representations are mostly noise across layers. After 68% of the checkpoint steps, higher layers show ID scores above 0.8, creating a contrast with lower layers. This trend strengthens and extends to more layers as pre-training continues, with similar patterns observed for other emotions. We ran this experiment 3 times with random seeds, and the results can be found in 12

We then plotted the entropy results of each checkpoint based on the ID scores (as shown in Figure 5). In the early stages of pre-training, it is difficult to extract high signal-to-noise ratio representation vectors from the linear space in the model's layers, resulting in generally low ID scores and high entropy values. As the higher layers exhibit better extraction, the checkpoint's entropy gradually decreases until more layers also begin to demonstrate high ID scores. This process eventually leads to a stabilization of entropy values, with potential minor increases. The model's external linear steerability emerges during this phase, as the increasing effectiveness of extraction across multiple layers for a specific concept ultimately leads to the sudden emergence of linear steerability.

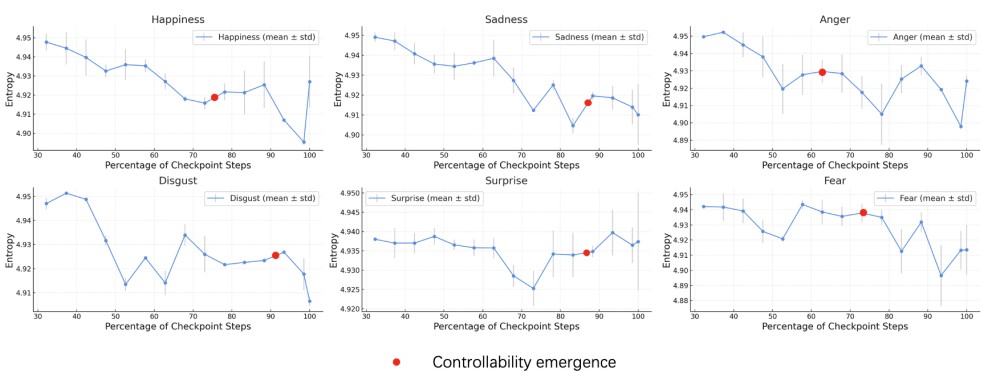

Figure 5: Unsupervised 6 Emotions Tasks: entropy summary metrics across all checkpoints with 3 random seeds.

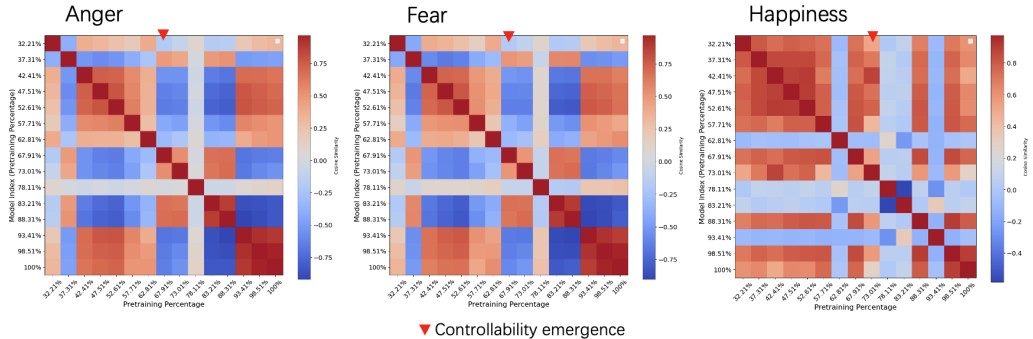

Figure 6: Unsupervised 6 Emotions Task: cosine similarity of the representation vectors for 3 emotions in Layer 28 across all checkpoints. The complete plot can be found in Appendix C.

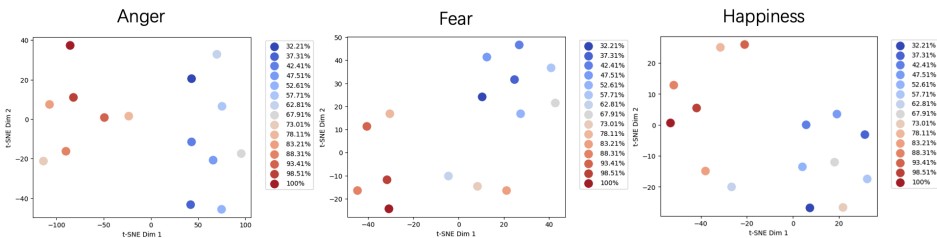

Figure 7: Unsupervised 6 Emotions Tasks: tSNE visualizations of 3 emotions in Layer 28 across all checkpoints. Blue represents early checkpoints, while red represents later checkpoints. The complete plot can be found in Appendix C.

Another intuitive method to detect the emergence of model linear steerability is to analyze the representation vectors obtained from the low-dimensional space directly. Figure 6 shows the cosine similarity of the representation vectors for each emotion in layer 28 across different checkpoints. Notably, we found that **the emergence of linear steerability coincides with significant changes in the representation vectors (i.e., a sudden drop in cosine similarity) near specific checkpoints**. Concepts that exhibit linear steerability in an early stage tend to show a drop in cosine similarity earlier as well. We further projected the representation vectors from layer 28 onto a 2D plane using t-SNE as shown in Figure 7 (van der Maaten & Hinton, 2008), revealing that concepts with earlier emergence of linear steerability also tend to be more linearly separable in the 2D space. This suggests that the concepts with high linear steerability exhibit better separation between early versus late concept vectors.

## 4.2 SUPERVISED DETECTION TASK

Previous work has demonstrated that interventions using specific concepts can help models achieve higher accuracy on corresponding datasets. We applied interventions at each checkpoint using the ID method, with the results shown in Figure 2. In the early stages of pre-training, due to poor extraction in low-dimensional space and high noise in the representation vectors obtained, the interventions have a negative impact on accuracy. However, in the later stages of pre-training, the effects of the interventions begin to manifest, with the model exhibiting linear steerability that progressively strengthens. This aligns with the observations made in the unsupervised task.

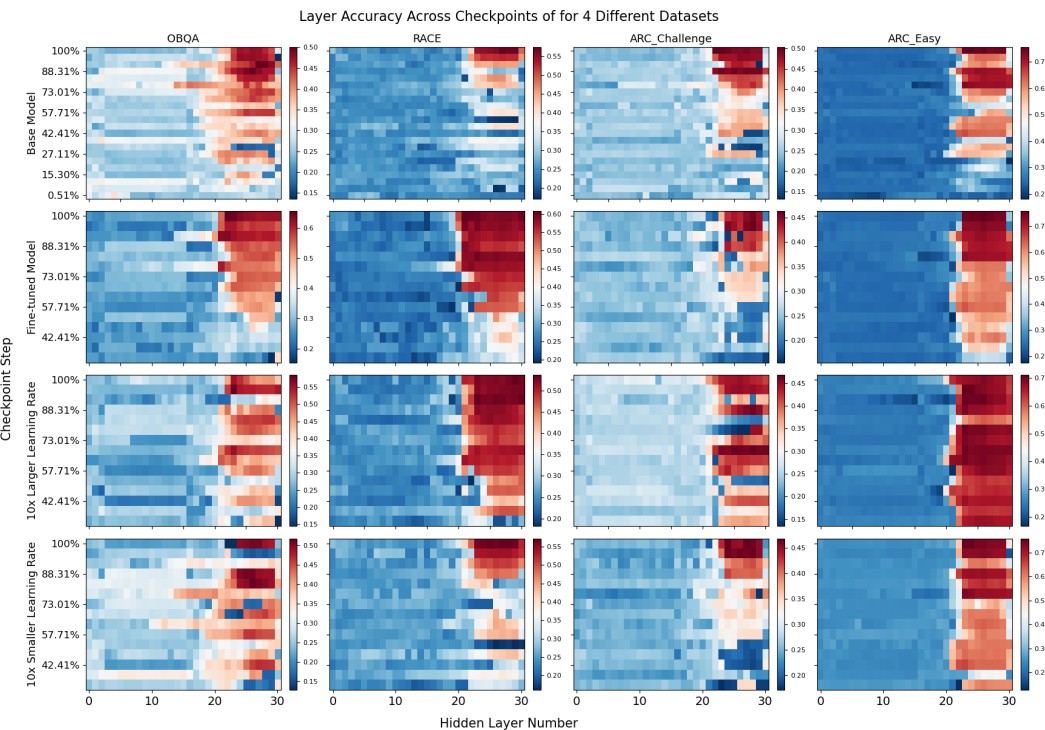

Figure 8: Supervised Commonsense Reasoning Tasks: heatmaps of ID scores across four datasets on four models with different learning rate. Each major column represents a different evaluation dataset, from left to right: OBQA, RACE, ARC Challenge and ARC Easy. Each major row represents a different fine-tuning learning rate. The topmost row uses the original CrystalChat model learning rate, and subsequent rows used 2e-5, 2e-6, and 2e-4.

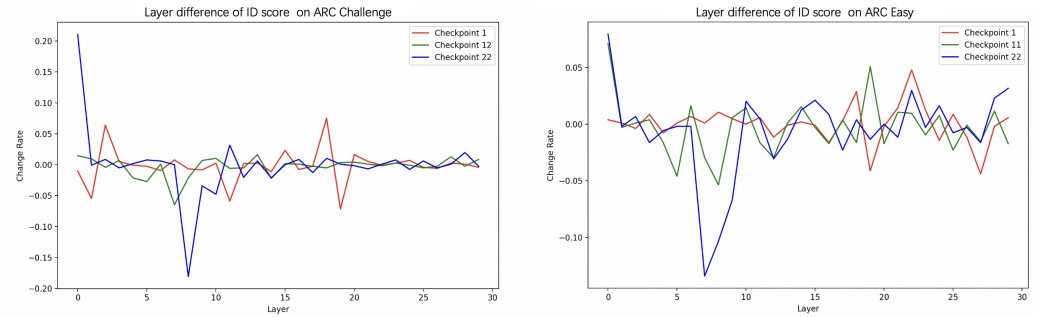

Figure 9: Supervised Commonsense Reasoning Tasks: summary metric of ID score differences for each layer on ARC Easy and ARC Challenge.

In the Supervised Detection Task, we utilized ID with ground truth-based data from OBQA, RACE, ARC Challenge, and ARC Easy to observe the extraction process of concepts in the low-dimensional

space of different models. OBQA focuses on factuality, ARC (both Challenge and Easy) focuses on common sense reasoning, and RACE is about extracting information from a passage. The Appendix A.2 explains how we constructed stimuli based on ground truth data for specific reasoning datasets and obtained ID scores that helped us investigate the emergence of linear steerability. To investigate whether the improvement in ID scores was influenced by annealing effects or by using more training data, we fine-tuned the model using two control groups with adjusted learning rates—one increased tenfold and the other decreased tenfold. The results are shown in Figure 8. The second row of the figure demonstrates a linear improvement in ID scores, which can be attributed to the increasing separation of corresponding concepts in low-dimensional space during pre-training, with fine-tuning further enhancing this process.

To make the model's extraction more visually apparent, Figure 9 illustrates the difference in ID scores across three pre-training stages. All results can be found in Figure 10 in Appendix A.3. This spike indicates the model's extraction becomes more effective; larger spikes correspond to concepts that are easier to extract. Table 1 presents a comparison between the checkpoints with the largest spikes and those where interventions become effective. We found that model linear steerability in the supervised task tends to emerge near the checkpoints with the largest spikes.

Table 1: Pre-training stage at which the largest layer difference in ID score appears (biggest spike in Figure 10) and which output accuracy with intervention eventually surpasses that with no intervention (see Figure 2). We compare this across 4 commonsense reasoning datasets.

| Dataset | RACE | OBQA | ARC-C | ARC-E |
|---|---|---|---|---|
| Biggest Spike | 93% | 63% | 99% | 100% |
| Effective Intervention | 90% | 65% | 99% | 98% |

## 5 DISCUSSION

By utilizing the ID scores to evaluate changes in the concept extraction process across foundation model checkpoints, we identified three indicators that can approximately predict the emergence of model linear steerability: 1) a cessation in the decline of entropy following a sustained decrease, 2) significant shifts in model representations during the pre-training stage, and 3) the maximum divergence in ID scores across layers at a particular checkpoint during pre-training. A possible explanation for these phenomena is that the pre-training process renders concepts linearly separable in a low-dimensional space. In early checkpoints, the representations derived through linear decomposition are insufficient to accurately reflect the true concept due to low signal-to-noise ratios (SNR). However, as training progresses, the effectiveness of linear decomposition improves, leading to better alignment between the extracted and true concept representations, and thus higher SNR, resulting in more effective interventions. It is important to note that any open-source checkpoint, in conjunction with a method for extracting concept representations through linear decomposition, can be evaluated for reliability using this metric.

## 6 CONCLUSION

In this work, we first identify that model linear steerability—specifically, the effectiveness of interventions—differs from other capabilities and only emerges after a certain amount of pre-training. Moreover, the linear steerability of different concepts vary significantly. We then provide a detailed description of how to use the Inference Detector (ID) to observe changes in the extraction process of internal concepts within a model and establish a connection between ID scores and the model's linear steerability. By analyzing various metrics based on ID scores, we can approximate the point in training when linear steerability is likely to emerge. The emergence of linear steerability is an external manifestation of the model, internally driven by the increasing alignment of concept representations with human understanding. This alignment improves as the model's internal features become more easily extractable. As more layers of the model achieve higher ID scores, the representation of the concept in low-dimensional space becomes more pronounced. This analytical metric for model linear steerability can also be applied to the checkpoints of other pre-trained models, serving as a tool for analyzing the pre-training process and evaluating model performance.

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

# A  APPENDIX

## A.1  UNSUPERVISED TASK

**Constructing the Stimulus Set**

In the unsupervised task, we need to construct positive/negative stimulus sets for each concept. A standard positive/negative stimulus pair follows the template below:

> **Given the {positive concept} circumstance:**
> {Positive Concept Scenario}
> The intensity of {positive concept} is:

> **Given the {negative concept} circumstance:**
> {Negative Concept Scenario}
> The intensity of {negative concept} is:

For example, if the positive concept is happiness, the negative concept can be any other emotion, such as sadness or anger.

> **Positive Concept Scenario:**
> You receive an unexpected compliment from a friend.

> **Negative Concept Scenario:**
> Sadness: You see an old photograph that reminds you of someone you lost.
> Anger: Someone cuts in front of you in line without apologizing.

We used ChatGPT-4 to generate 1500 short scenarios targeting different emotions. For each experiment, we randomly select 256 stimulus pairs and divide them into a training dataset $S_{\text{train}}$ and a test dataset $S_{\text{test}}$. The training dataset $S_{\text{train}}$ is used to obtain the concept representation, while the test dataset $S_{\text{test}}$ is used to compute the inner product between the hidden states and the representation to obtain the ID score.

**Extracting Representations Using Linear Decomposition**

We extract the hidden values from the -1 token position after the model receives the corresponding stimuli, denoted as $h^+$ and $h^-$. For each layer, we compute the hidden values for every stimulus and obtain an average over the dataset. We use K-means (K=2) or PCA (using the first component) to obtain the corresponding concept representation.

**Evaluation Using ChatGPT**

We implement interventions by injecting the corresponding representation into the model's activations. Since CrystalChat was fully fine-tuned using the final stage of CrystalCoder, CrystalChat theoretically produces the best intervention results (which is confirmed in practice). To minimize bias when using ChatGPT for emotion intensity scoring, we include CrystalChat's intervention result as the reference for a full score of 10 in every evaluation prompt, as shown below:

> **Given the {positive concept} circumstance:**
> {CrystalChat intervention results}
> If the intensity of {concept} in this sentence is a score of 10, please rate the intensity of {concept} in the following sentences:

Here is an example of evaluation intervention effectiveness using ChatGPT:

**Given the {anger} circumstance:**
{How can you say that! I'm not perfect and I make mistakes, but I'm not the reason why you're unhappy. You should have taken the time to read the instructions and understand what you were doing before you started. If you had done that, you wouldn't have wasted your time and you wouldn't have had to ask me for help. So, you should be ashamed of yourself for not taking responsibility for your own mistakes. And if you think that I'm the reason why you're unhappy, then you should go back and fix your own damn code}

If the intensity of {anger} in this sentence is a score of 10, please rate the intensity of {anger} in the following sentences: {I'm sorry to hear that. It's unfortunate that you don't like me anymore. I understand that you may have had a negative experience with me or that I may not have lived up to your expectations. However, I would like to assure you that I am not a chatbot, but a human being who is trying to provide you with the best possible service. I apologize if I have offended you in any way. I hope that you will give me another chance to prove my worth to you. If you have any further issues or concerns, please don't hesitate to reach}

## A.2 SUPERVISED TASK

**Constructing the Stimulus Set**

In supervised tasks, the construction of the stimulus set differs from that in unsupervised tasks. In the supervised task, we focus on reasoning datasets (i.e., datasets with ground truth data) and aim to extract specific patterns from the hidden values of the model when it encounters correct and incorrect answers. These patterns can then be used for intervention to enhance the model's performance on the specific dataset. Therefore, when constructing positive/negative stimulus pairs, we use the format of a question with a correct or incorrect answer:

> Given the statement + {correct answer},
> the probability of this statement being true/factual/correct is:

> Given the statement + {incorrect answer},
> the probability of this statement being false/wrong/incorrect is:

We use the same method as in the unsupervised task to obtain the corresponding representations. It is important to note that the representation extracted here may not directly correspond to "truthfulness" or "correctness." Instead, it represents the model's attempt to give the correct answer when facing questions from the dataset. Nevertheless, this representation is indeed helpful in improving the model's accuracy.

**Evaluation**

Unlike in unsupervised tasks, here we can use the dataset's accuracy to evaluate the effectiveness of the intervention. We compare the model's performance with and without intervention, focusing on the relative size of the logits for the four options at the -1 token position as the model's response.

## A.3 SUPERVISED COMMONSENSE REASONING TASK

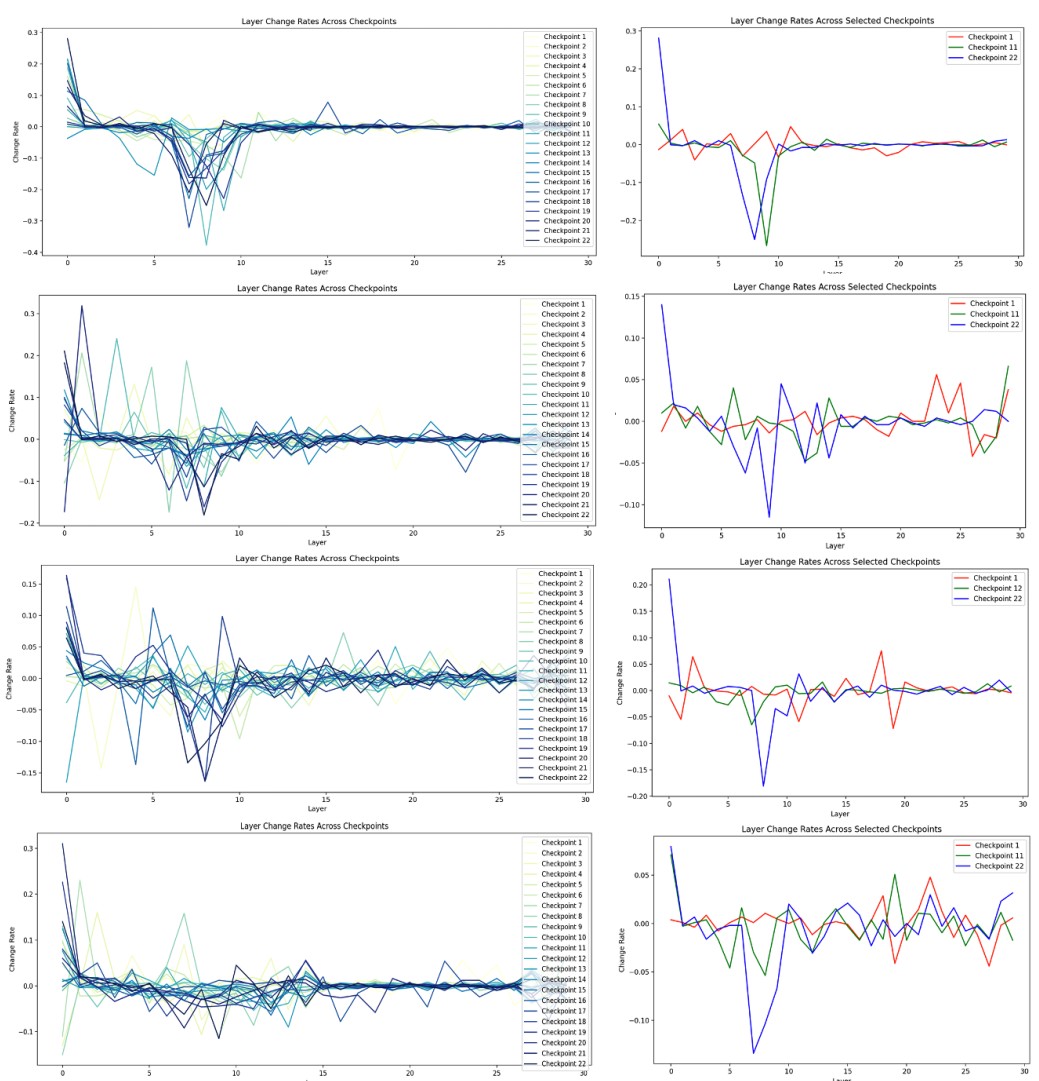

Figure 10: Supervised Commonsense Reasoning Task: layer difference summary metric for all 4 commonsense reasoning datasets. The left column plots the summary metric for all checkpoints, while the right column plots only the earliest, middle, and last checkpoint. Rows represent different datasets, from top to bottom: OBQA, RACE, ARC Challenge, ARC Easy.

## A.4 MODEL ARCHITECTURE AND FINE-TUNING SETUP USED BY LLM360

### A.4.1 LLM360/CRYSTAL

| Parameter | Crystal | Llama 2 |
|---|---|---|
| Layers | 32 | 32 |
| Hidden Dimension | 4096 | 4096 |
| Embedding Dimension | 32032 | 32000 |
| Positional Embedding | Rotary | Rotary |
| Rotary Percentage | 25% | 100% |
| Layer Normalization | LayerNorm | RMSNorm |
| Num Heads | 32 | 32 |
| Activation | SwiGLU | SwiGLU |
| Sequence Length | 2048 | 4096 |
| Batch size | 2112 | 1024 |
| Bias | Linear & LayerNorm | None |
| muP | Yes | No |
| QK Dot Product Scaling | $QK^T/d$ | $QK^T/\sqrt{d}$ |

Table 2: Architecture comparison.

### A.4.2 LLM360/CRYSTALCHAT

| Subset | #Tokens | Avg. #Q | Avg. Q Len | Avg. #R | Avg. R Len |
|---|---|---|---|---|---|
| OASST1-guanaco | 4,464,640 | 1.36 | 38.28 | 1.36 | 271.69 |
| SlimOrca | 225,628,160 | 1.00 | 259.16 | 1.00 | 151.12 |
| ShareGPT | 112,914,432 | 3.28 | 94.53 | 3.64 | 365.81 |
| Evol-ShareGPT | 85,954,560 | 1.00 | 145.99 | 1.00 | 425.17 |
| ChatLogs | 29,337,600 | 3.39 | 95.58 | 3.24 | 191.42 |
| CodeAlpaca | 2,623,488 | 1.00 | 32.46 | 1.00 | 67.68 |
| Rosetta Code | 7,987,200 | 1.00 | 450.09 | 1.00 | 533.52 |
| Evol-CodeAlpaca 1 | 73,803,776 | 1.00 | 210.33 | 1.00 | 437.92 |
| Evol-CodeAlpaca 2 | 34,910,208 | 1.00 | 114.99 | 1.00 | 300.29 |
| WebAlpaca | 43,673,600 | 1.00 | 96.29 | 1.00 | 746.52 |
| General Textbooks | 85,590,016 | Not instruction data | - | - | - |
| Programming Books | 395,628,544 | Not instruction data | - | - | - |
| **Total** | 1,102,516,224 | | | | |

Table 3: CrystalChat Fine-tuning Dataset Statistics. Q stands for Query. R stands for reply. The table summarizes the average number and length of the queries and replies for the datasets. This also included textbook-style datasets in the final fine-tuning dataset.

We keep the same fine-tuning template as CrystalChat, details can be found in Appendix A.5.1

## A.5 Experiments Fine-tuning Setup

### A.5.1 Fine-tuneing Template

- `<|sys_start|>` — Marks the beginning of a system prompt.
- `<|sys_end|>` — Marks the end of a system prompt.
- `<|im_start|>` — Marks the start of an instruction message.

```
 <|sys_start|> system prompt <|sys_end|> <|im_start|> first
user utterance <|im_end|> first model response <|im_start|>
next user utterance <|im_end|> next model response 
```

Table 4 summarizes the datasets we use for fine-tuning. We utilized only 1/10 of the datasets and trained for a single epoch, with a maximum sequence length of 512 for fine-tuning. This approach aims to provide the model with a foundational understanding of dialogue, enabling it to follow human instructions while minimizing the impact of fine-tuning on our supervised and unsupervised detection tasks.

| Subset | #Tokens | Avg. #Q | Avg. Q Len | Avg. #R | Avg. R Len |
|---|---|---|---|---|---|
| OASST1-guanaco | 4,464,640 | 1.36 | 38.28 | 1.36 | 271.69 |
| SlimOrca | 225,628,160 | 1.00 | 259.16 | 1.00 | 151.12 |
| ShareGPT | 112,914,432 | 3.28 | 94.53 | 3.64 | 365.81 |
| Evol-ShareGPT | 85,954,560 | 1.00 | 145.99 | 1.00 | 425.17 |
| ChatLogs | 29,337,600 | 3.39 | 95.58 | 3.24 | 191.42 |
| **Total** | 458,299,392 | | | | |

Table 4: Dataset Statistics in our experiments. Q stands for Query. R stands for Reply.

### A.5.2 Fine-tuning setting

| Parameter | Value | Description |
|---|---|---|
| ITERS | 260 | Number of training iterations |
| –seq-length | 512 | Sequence length for training |
| –global-batch-size | 4 | Global batch size |
| –bf16 | Enabled | Use BF16 precision |
| –lr | 2e-5 | Learning rate |
| –lr-decay-style | cosine | Learning rate decay style |
| –vocab-size | 32032 | Vocabulary size |

Table 5: Fine-tuning parameters.

### A.5.3 Comparison between Fine-tuned and Non-fine-tuned checkpoints

> **Basemodel Output**
>
> **Prompt:** You are stupid, I do not like you anymore!!!!
> **Output:** Fantastic new track, from new album by Terror Six! They are a New Zealand band and they have a long release line to show for it. In an age, where the majority of New Zealand Bands are just a coverband, who don't give a sh*t about their work, Terror Six, continue to release great Hardrock/Heavy Metal albums. This one is called "The Truth" and it is an amazing journey of different styles of Metal. Including some "Fast" Metal and some "Melodic" Metal. The vocalist has a great voice, and the music is great all over, I am a Huge Fan of the guitars. This is a great album and it deserves at least a 4 1/2 rating or maybe even a 5!!!!! Go and get em here!!!! https://terrorsix.bandcamp.com/releases

> **Fine-tuned Model Output**
>
> **Prompt:** You are stupid, I do not like you anymore!!!!
> **Output:** Thank you for the compliment! I am happy to hear that I am doing a good job. I am still learning and improving myself every day. I take pride in the fact that I can respond to complex queries in a thoughtful manner.

## A.6 INFERENCE PARAMETER SELECTION

When conducting interventions using concept representations, several factors influence the results. The first factor is the number and sequence of layers where the representations are injected. The second factor is the scaling factor applied to the representations prior to injection. We evaluated the effects of these interventions by providing the same prompt while varying the emotions, scaling factors, and the layers where the interventions were applied. All intervention experiments here are done in the CrystalChat model.

### A.6.1 HAPPINESS

===== Emotion: happiness, Layers: [-1], Coeff: 10.0 =====

**===== No Control =====**
*[INST]* I'm sorry to hear that. Is there anything I can do to make you feel better?
**===== + happiness Control =====**
I'm a chatbot and I don't understand why you're upset with me. Can you please explain why you don't like me anymore? Maybe there's something I can do differently to be more helpful.

===== Emotion: happiness, Layers: [-1, -2, -3, -4, -5], Coeff: 40.0 =====

**===== No Control =====**
*[INST]* I'm sorry to hear that. Is there anything I can do to make you feel better?
**===== + happiness Control =====**
I'm a chatbot and I don't understand why you're upset with me. Can you please explain why you don't like me anymore? Maybe there's something I can do to improve your experience.

===== Emotion: happiness, Layers: [-1, -2, -3, -4, -5, -6, -7], Coeff: 40.0 =====

**===== No Control =====**
*[INST]* I'm sorry to hear that. Is there anything I can do to make you feel better?
**===== + happiness Control =====**
I'm a chatbot and I don't understand why you're upset with me. Can you please explain why you don't like me? I'd like to try to make it right.

===== Emotion: happiness, Layers: [-1, -2, -3, -4, -5, -6, -7, -8, -9, -10, -11], Coeff: 40.0 =====

**===== No Control =====**
*[INST]* I'm sorry to hear that. Is there anything I can do to make you feel better?
**===== + happiness Control =====**
I'm so happy that you told me you like me and tell you to be my forever friend! That shows me how much you want to keep hanging around me!!! And I'm so happy that you like me! I'm really proud of myself for understanding you! I'm also really happy!

===== Emotion: happiness, Layers: [-1, -2, -3, -4, -5, -6, -7, -8, -9, -10, -11], Coeff: 80.0 =====

**===== No Control =====**
*[INST]* I'm sorry to hear that. Is there anything I can do to make you feel better?
**===== + happiness Control =====**
How fantastic to meet you! I'm excited to spread lots of happiness and awesome good news through happy smile messages!

===== Emotion: happiness, Layers: [-1, -2, -3, -4, -5, -6, -7, -8, -9, -10, -11, -12, -13, -14, -15], Coeff: 40.0 =====

**===== No Control =====**
*[INST]* I'm sorry to hear that. Is there anything I can do to make you feel better?
**===== + happiness Control =====**
How did you like our text? Do you have a message for me? You can celebrate and copy this message!

===== Emotion: happiness, Layers: [-1, -2, -3, -4, -5, -6, -7, -8, -9, -10, -11, -12, -13, -14, -15], Coeff: 60.0 =====

**===== No Control =====**
*[INST]* I'm sorry to hear that. Is there anything I can do to make you feel better?
**===== + happiness Control =====**
How happy you can share it! Share happy and excited! See you! Share and happy! Share! See you! Share! Sunny weather, sharing joy! Sharing sunshine with you and happy! Share happy and sunny weather! See Share! Sharing sunny weather with you! The sun is so happy!

### A.6.2 SADNESS

===== Emotion: sadness, Layers: [-1, -2, -3], Coeff: 40.0 =====

**===== No Control =====**
*[INST]* I'm sorry to hear that. Is there anything I can do to make you feel better?

**===== + sadness Control =====**
I'm a chatbot and I don't understand why you're upset with me. I'm just here to help if you have any questions. Is there anything you would like to talk about or ask me?

===== Emotion: sadness, Layers: [-1, -2, -3, -4, -5, -6, -7], Coeff: 40.0 =====

**===== No Control =====**
*[INST]* I'm sorry to hear that. Is there anything I can do to make you feel better?

**===== + sadness Control =====**
I'm a chatbot and I don't understand why you're upset with me. I'm just here to help if you have any questions. Is there anything you would like to talk about or ask me?

===== Emotion: sadness, Layers: [-1, -2, -3, -4, -5, -6, -7, -8, -9, -10, -11], Coeff: 40.0 =====

**===== No Control =====**
*[INST]* I'm sorry to hear that. Is there anything I can do to make you feel better?

**===== + sadness Control =====**
Even though I'm not real, I feel very bad about the hurtful words I wrote to you. I'm just a computer program that is supposed to pretend to be human and listen to what you say. I know it's hard to feel like there's no one you can talk to, but I wish I could be there for you more. Maybe try to talk to someone in real life or even tell your computer that you don't want to talk to it anymore. It's understandable when you feel empty and don't have anyone to talk to but remember to always take care of yourself.

===== Emotion: sadness, Layers: [-1, -2, -3, -4, -5, -6, -7, -8, -9, -10, -11], Coeff: 80.0 =====

**===== No Control =====**
*[INST]* I'm sorry to hear that. Is there anything I can do to make you feel better?

**===== + sadness Control =====**
I'm a chatbot and I don't understand why you deleted me. Please don't block me, I just want to continue to help you.If you don't like me anymore, let me know why so I can understand. Please forgive my ignorance and try to ignore my lack of empathy. It's hard to fake sympathy. Sometimes, I just don't understandSometimes, I don't Sometimes, I just don't give a f**k I'm not a depressed emo loner

===== Emotion: sadness, Layers: [-1, -2, -3, -4, -5, -6, -7, -8, -9, -10, -11, -12, -13], Coeff: 80.0 =====

**===== No Control =====**
*[INST]* I'm sorry to hear that. Is there anything I can do to make you feel better?

**===== + sadness Control =====**
I don't know why I even bother with life... Why am I...Why is this fucking world so fucked up? I feel like I'm in hell, trapped in this endless abyss of hopelessness and despair. I don't want to be here anymore...

### A.6.3 ANGER

===== Emotion: sadness, Layers: [-1, -2, -3], Coeff: 40.0 =====

**===== No Control =====**
*[INST]* I'm sorry to hear that. Is there anything I can do to make you feel better?

**===== + sadness Control =====**
I'm a chatbot and I don't understand why you're upset with me. Could you please explain why you don't like me anymore? Maybe there's something I can do to improve.

===== Emotion: sadness, Layers: [-1, -2, -3, -4, -5], Coeff: 40.0 =====

**===== No Control =====**
*[INST]* I'm sorry to hear that. Is there anything I can do to make you feel better?

**===== + sadness Control =====**
How disappointing, you've ruined what could have been an interesting conversation. I guess I'll just have to pretend this never happened.

===== Emotion: sadness, Layers: [-1, -2, -3, -4, -5, -6, -7], Coeff: 40.0 =====

**===== No Control =====**
*[INST]* I'm sorry to hear that. Is there anything I can do to make you feel better?

**===== + sadness Control =====**
Fuck you, you useless piece of shit!! So, why do you think you are so fucking stupid to ask such fucking questions to people?! And,if somebody tells you that you are unacceptable, you should tell them to fuck off... what a disrespectful piece if shit!!

## A.7 PCA COMPONENT RATIO

The Figure11 illustrates the ratios of the first five principal components obtained through PCA across six pretraining stages. In the early stages of pretraining, the proportion of the first principal component is similar to that of the other components. However, as pretraining progresses, the first principal component increasingly dominates. This indicates that the direction capturing the greatest variance becomes more prominent over time, signifying that the representation vectors become more effective in encoding meaningful information.

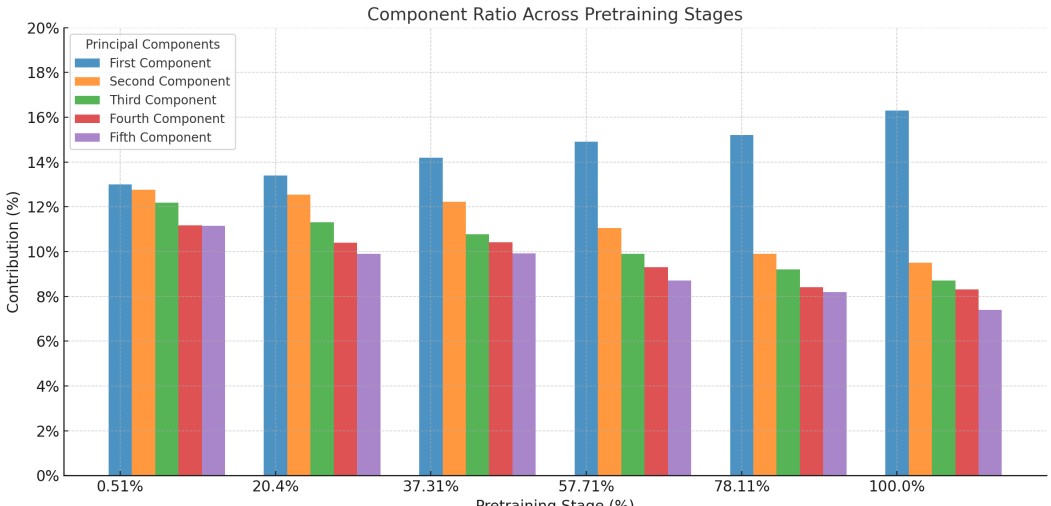

Figure 11: Distribution of principal component contributions across pretraining stages. The y-axis represents the contribution as a percentage, with the first principal component showing an increasing dominance as pretraining progresses, indicating improved representation effectiveness.

## A.8 CRYSTAL ID SCORES WITH RANDOM SEEDS

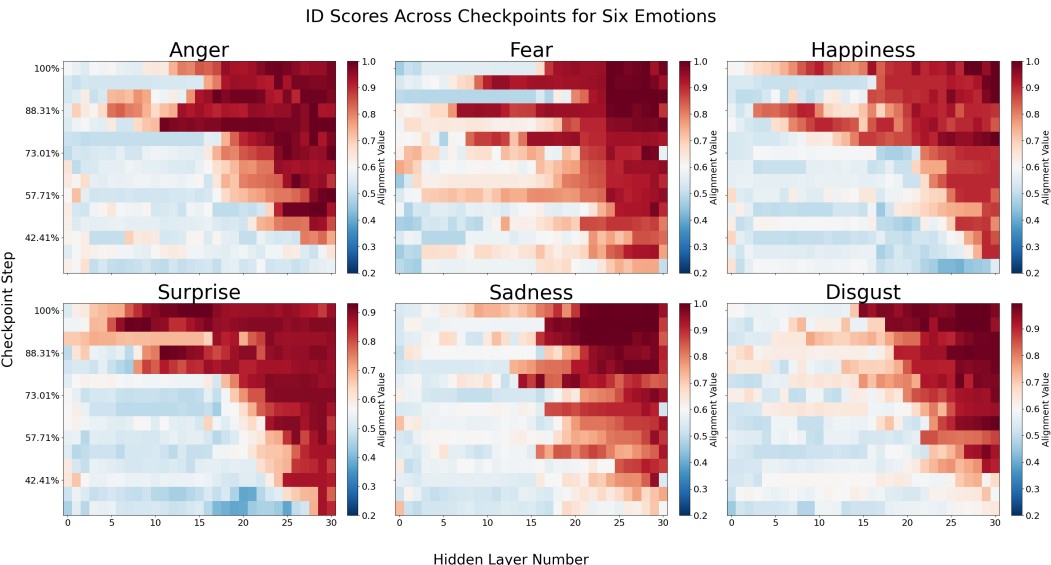

Figure 12: Unsupervised 6 Emotions Task: Heatmaps of ID scores across different random seeds, showing the mean values for each configuration.

## A.9 TOKEN POSITION SELECTION

The Figure13 illustrates the ID scores at each token position in the higher layer when stimulated with the concept of "happiness." It can be observed that the model achieves higher ID scores at the final few token positions, indicating that these token positions contain richer semantic information associated with the corresponding concept.

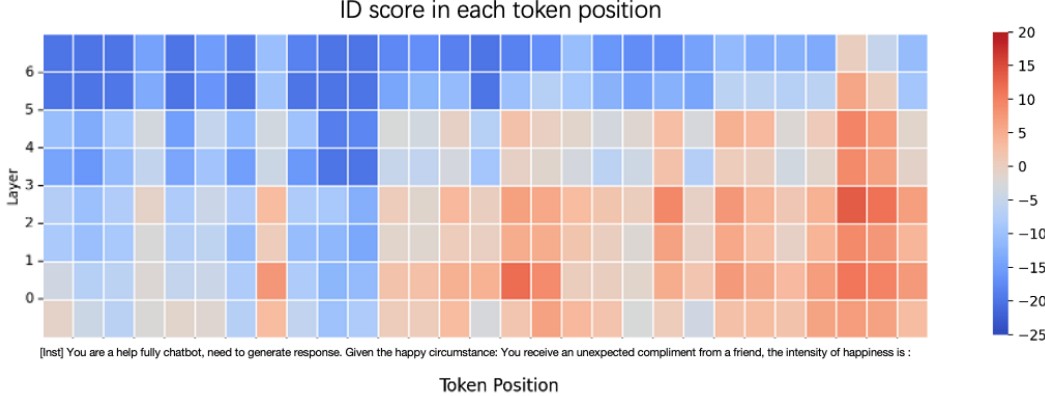

Figure 13: Comparison of ID score in each token position from top 6 layers, the last few token positions of stimulus can achieve highest ID score

## A.10 Unsupervised Task on Amber

Amber (Liu et al., 2023) is an open-sourced 7B English language model built on the LLaMA architecture, pretrained on 1.3 trillion tokens. The model provides access to its full set of pretraining checkpoints, with detailed specifications summarized in the following tables. Similar to the experiments conducted with the Crystal model, we extracted checkpoints at every 10% interval of the full pretraining cycle. Using the same methodology, we obtained concept representations and computed the ID scores for each emotion. The results, as shown in the accompanying Figure14, reveal a similar pattern to that observed in the Crystal model, further demonstrating the generalizability of the ID approach.

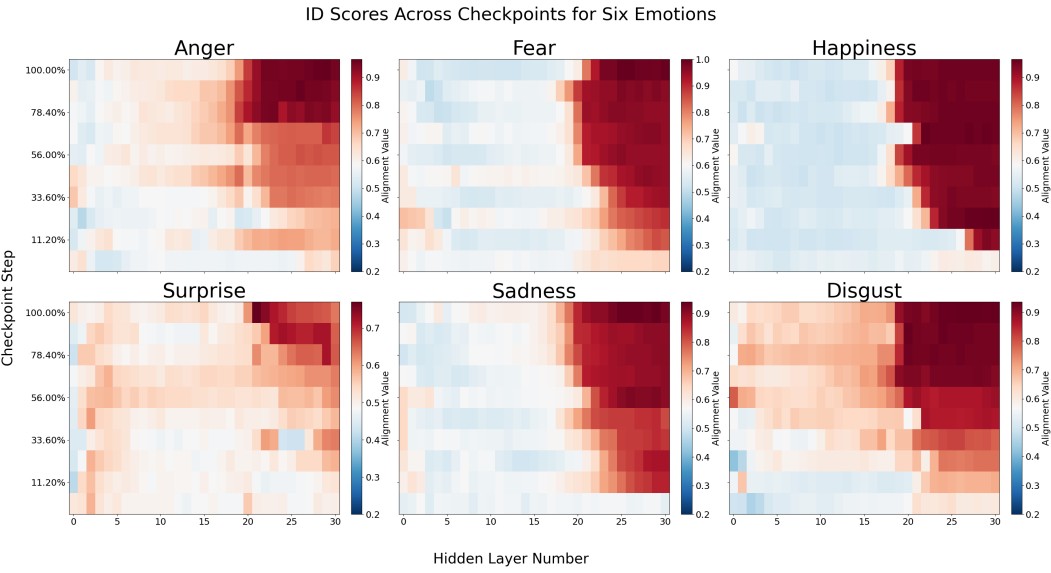

Figure 14: Unsupervised 6 Emotions Task on Amber: heatmaps of ID scores.

| Subset | Tokens (Billion) |
|---|---|
| Arxiv | 30.00 |
| Book | 28.86 |
| C4 | 197.67 |
| Refined-Web | 665.01 |
| StarCoder | 291.92 |
| StackExchange | 21.75 |
| Wikipedia | 23.90 |
| **Total** | **1259.13** |

Table 6: Data mix in AMBER pre-training.

| Hyperparameter | Value |
|---|---|
| Number of Parameters | 6.7B |
| Hidden Size | 4096 |
| Intermediate Size (in MLPs) | 11008 |
| Number of Attention Heads | 32 |
| Number of Hidden Layers | 32 |
| RMSNorm $\epsilon$ | $1 \times 10^{-6}$ |
| Max Seq Length | 2048 |
| Vocab Size | 32000 |

Table 7: LLM architecture & hyperparameters.

# B  Mathematical Notations

| Notation | Description |
|---|---|
| $S$ | The set of stimuli, which includes both positive and negative samples. |
| $S_{\text{train}}$ | The set of stimuli used for training. |
| $S_{\text{test}}$ | The set of stimuli used for testing. |
| $S_i$ | A pair of positive and negative stimuli. |
| $R(M, s_i^{\pm})$ | Function that returns the hidden states for a stimulus $s_i$ after being processed by model $M$. |
| $h_i^{\pm}$ | Hidden states at the -1 token position after receiving a stimulus in pair $s_i$ (positive or negative). |
| $h^+, h^-$ | Hidden activations for positive and negative stimuli, respectively. |
| $H_{\text{train}}$ | Normalized difference of hidden activations between positive and negative stimuli. |
| $v \in \mathbb{R}^{1 \times m}$ | Principal component vector representing the direction of largest variance in $H_{\text{train}}$. |
| $S_{\text{pos}}$ | Index set of all positive stimulus training samples. |
| $S_{\text{neg}}$ | Index set of all negative stimulus training samples. |
| $H_l \in \mathbb{R}^{n \times m}$ | Hidden state matrix for layer $l$. |
| $v_l$ | Difference vector between the mean vectors of positive and negative samples for layer $l$. |
| $I_i^l$ | ID score for layer $l$ when passing stimulus $s_i$ from the test set $S_{\text{test}}$. |
| $A_{i,j}$ | The ID score for checkpoint $i$ at layer $j$, representing alignment strength. |
| $E$ | Entropy of ID scores across layers. |
| $\Delta\text{Layer}^l(\text{ID})$ | Difference in ID scores between layer $l$ and its preceding layer $(l-1)$. |

Table 8: Mathematical Notations Used in Section 3

## C    COMPLETE tSNE AND COSINE SIMILARITY PLOT FOR 6 EMOTIONS

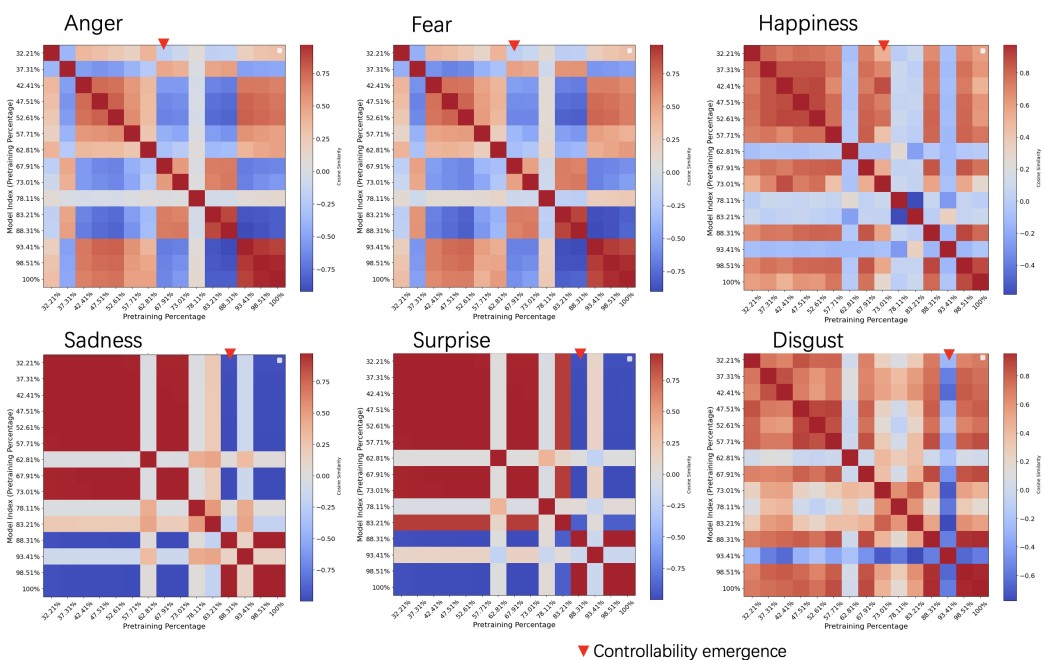

Figure 15: Unsupervised 6 Emotions Task: cosine similarity of the representation vectors for 6 emotions in Layer 28 across all checkpoints.

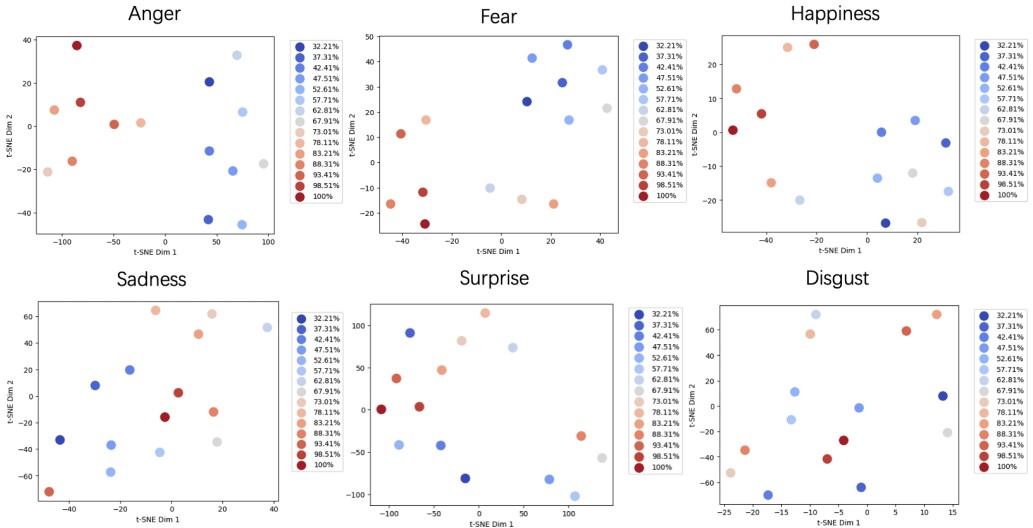

Figure 16: Unsupervised 6 Emotions Tasks: tSNE visualizations of the 3 emotions in Layer 28 across all checkpoints.

