# OpenReview forum: "How does controllability emerge in language models during pretraining?"
_ICLR.cc/2025/Conference — Submitted to ICLR 2025_

### Official Review · Reviewer_6yEU · 2024-10-27

**Soundness:** 3
**Presentation:** 2
**Contribution:** 3
**Rating:** 6
**Confidence:** 4

**Summary:**

The paper explores how controllability emerges during the pretraining process of language models. The authors propose a novel framework, named Intervention Detector (ID), which applies dimensionality reduction techniques (like PCA) to the hidden states of models under different stimuli, allowing for the extraction of concept-specific representations. This method allows for targeted interventions, helping models respond more predictably to specific prompts.
The paper finds that controllability in language models is an emergent property that develops as pre-training progresses.

**Strengths:**

1. The idea of investigating the concept of controllability within pre-trained language models is novel and interesting.
2. The paper has been experimented with fairly extensively, and the results show good potential.

**Weaknesses:**

The paper lacks clarity and most of the practical details and its design principles remain vague or unresolved. Please see the Questions.

**Questions:**

1. I think there are some operations that you need to explain why:
- In Hidden States Collection, why do you collect the hidden states at the -1 token position?
- In Dimensionality Reduction, why do you compute the difference of hidden activations and compute PCA? How is this related to the latent concept?
- Where is the definition of $A_i$ in Analyzing ID scores Across Layers?

2. Have you considered other methods of selecting stimuli? For example, based on perplexity.

3. You just randomly selected 256 stimuli, which is heavily influenced by the random seed, so I think you need to run different tests to see how different seeds affect the random selection.

---

> ### Author Response · Authors · 2024-11-13
>
> Thank you very much for your thoughtful review and valuable feedback,
>
> We appreciate that you pointed out the opaque in our experiment operations, we notice that those caused a problem in understanding how we recognize the capability of intervention. We will give more explanations of our experiment operations, and use random seeds to select stimuli.
>
> We hope that incoming revisions will address your concerns.

---

> ### Author Response · Authors · 2024-11-22
>
> Thank you very much for your thoughtful review.
>
> Regarding the rationale for extracting hidden states at the -1 token position, we conducted an experiment to observe the performance of ID scores at different token positions in the top layer. The results are presented in Appendix A.9 (page 25). For specific stimuli, higher ID scores indicate better alignment between the concept vector and the model's hidden states. We found that the hidden states corresponding to the final tokens of the stimuli tend to achieve higher ID scores. Therefore, we chose the -1 token position for our analysis. Additionally, we utilized the hidden states at the final token position because this position typically summarizes all preceding context, effectively capturing the model’s final representation of the entire input. We have added these clarifications in lines 237 to 241 of the manuscript.
>
> In the process of applying PCA, our goal was to ensure that opposing concepts become the primary factor influencing the model's response. PCA captures the direction of maximum variance in the data, which aligns with our experimental design: the hidden states of the model are expected to encode distinct and opposing concepts, such as happiness and sadness, which can then be captured as specific directions by PCA. Intuitively, the vector obtained at this stage represents the model’s understanding of the corresponding concept. In Appendix A.7 (page 24, lines 1250–1270), we demonstrate how the ratio of each PCA component evolves over different pretraining stages. Toward the later stages of pretraining, the first component becomes increasingly dominant, and this component effectively enables interventions. This leads us to conclude that the vector derived through PCA is associated with a specific concept.
>
> Our intervention experiments, presented in Appendix 6.1 (page 21), further validate that the concept representations obtained are consistent with human understanding of those concepts.
>
> We apologize for the lack of clarity in the "Analyzing ID Scores Across Layers" section. To address this, we have added details about the entropy calculation in lines 304 to 313 of the revised manuscript. Additionally, we have included a table (Appendix B, page 27) to define all the notations used in Section 3.
>
> Our stimuli were generated using ChatGPT-4. Specifically, we created 1,500 short scenarios targeting six emotions. These scenarios were carefully reviewed, and irrelevant ones were manually removed. The selection criteria were based on human judgment, as our goal was to ensure that the stimuli represented distinct concepts that align with human understanding. While we considered alternative methods for selecting stimuli, we determined that human judgment was more reliable in this case, as it allows us to steer the concepts in directions that are interpretable and meaningful.
>
> We acknowledge that our experiments may be affected by random seeds. To address this, we conducted three independent runs for ID scores (Appendix A.8, page 24) and entropy plots (page 8, lines 378–388), and five independent runs for the supervised task intervention experiments (page 3, lines 108–123). Error bars are included in the respective plots to account for variability.
>
> We sincerely appreciate your feedback and have made these updates to improve the clarity and rigor of our manuscript. Thank you once again for your valuable comments.

---

> > ### Comment · Reviewer_6yEU · 2024-11-25
> > **Official Comment**
> >
> > I have carefully read the reviews from other reviewers and responses from the authors. Thanks for the detailed responses from the authors, I raise my rating to 6

---

> > > ### Author Response · Authors · 2024-11-25
> > >
> > > Thank you for taking the time to carefully review our work and for your thoughtful comments. We greatly appreciate your recognition of our efforts in addressing the concerns raised during the review process. We are glad that our clarifications were helpful and are grateful for your updated rating :)

---

> > > ### Author Response · Authors · 2024-12-03
> > >
> > > Thank you once again for your thorough review of our paper and for providing valuable suggestions. Aside from some formal errors and concerns for which results cannot be obtained in the short term, we have attempted to address most of the concerns raised by the other reviewers. We hope that the final revised paper and the ongoing discussion will more clearly convey our work and address your concerns. Thank you!

---

### Official Review · Reviewer_Jm63 · 2024-11-01

**Soundness:** 2
**Presentation:** 2
**Contribution:** 2
**Rating:** 5
**Confidence:** 3

**Summary:**

This paper explores when controllability (via representation steering) emerges in language model pretraining. To do so, the authors devise the Intervention Detector (ID) method, which, given a set of layer representations, identifies when a certain concept becomes linearly encoded. They find that controllability, for several concepts, emerges suddenly during pre-training.

**Strengths:**

The idea to explore the emergence of linear steerability over pre-training is quite novel; past works only focus on the final configuration of the LM (but this is for good reason, see weaknesses). Experiments test a broad range of concepts.

**Weaknesses:**

I have several concerns about the paper, summarized broadly as follows:

**Major weaknesses (impacted score)**
1. __Motivation unclear/unconvincing:__ it was unclear why when controllability emerges should matter. In particular,
    - While the authors state that past work focuses on steerability of already-trained language models, this is the actual use case of LM steering. In contrast, the authors would need to make a strong case for investigating the emergence of controllability-- to me, it is unclear why one cares about representation steering on a not-fully-trained model, as the model itself would never be deployed. I would suggest to clarify the context in which emergence of controllability is useful.
    - Controllability in and of itself doesn't mean much: what does it mean for concept representations to emerge "early in pretraining" when the LM presumably has not converged to a good distribution of language?

2. __Imprecise definitions:__ the paper is about ``controllability" at large, yet only addresses a quite narrow domain within _linear_ control, adding an unscaled input to the representation. Controllability is a broad term that spans also nonlinear control-- I would change the emphasis to just additive representation steering.

3. __Methodological issues__:
    - Even though the authors state it is out of scope, it is important to test on another LM family for generalizability.
    - Did you try different scaling factors? In the literature, the scaling strength is crucial for performance [1-3]. If different scaling factors were not tried, could the authors please provide a justification?
    - Experiments are ideally run with several random seeds. For instance, it is hard to gauge the true effect for ARC Challenge and ARC Easy without error bars-- for OBQA, the gap is also quite small. Could the authors please provide error bars with the plots?

**Minor weaknesses (didn't impact score)**
1. Missing citations: https://arxiv.org/abs/2310.04444v3 https://arxiv.org/abs/2405.15454
2. l205: summarize the RepE method (e.g., are you referring to RepE linear combination, piece-wise operation, or projection?)
3. l210: the algorithm used for the Unsupervised task is unclear. What does it mean to obtain hidden values? If you mean the hidden states of the LM, which state are we using (e.g., last token)?

[1] Turner et al. 2023 https://arxiv.org/abs/2308.10248

[2] Li et al. 2023 https://arxiv.org/abs/2306.03341

Missing references:

[3] Cheng et al. 2024 https://arxiv.org/abs/2405.15454

[4] Soatto et al. 2023 https://arxiv.org/abs/2305.18449

[5] Bhargava et al. 2024 https://arxiv.org/abs/2310.04444v3

**Questions:**

See weaknesses

---

> ### Author Response · Authors · 2024-11-13
>
> Thank you very much for your thoughtful review and valuable feedback. We are committed to addressing your concerns with the following actions:
>
> Clarify Our Motivation: Thank you for pointing out that our motivation is not clearly conveyed in the paper. With the slowing progress of scaling laws, the research focus has shifted more towards test time compute efficiency rather than simply continuing pre-training or fine-tuning. As shown in Fig. 1, each concept may exhibit different levels of performance during pre-training. Even if we stop pre-training at a certain stage, some concepts may still have room to improve their "steerability." This knowledge allows us to make pre-training more efficient by targeting specific concepts and determining the optimal stopping point, even when the loss has already converged. We will ensure that this motivation is clearly explained in the revised version.
>
> Clarify the Definition of Controllability: We sincerely appreciate your concern regarding the term "controllability." We realize that it may lead to multiple interpretations and potential misunderstandings. We agree that the term may not be fully accurate in describing our findings, and we will consider using more precise alternatives, such as "editability" or "linear steerability," in the revised version.
>
> Methodological Issues: We acknowledge the lack of generalizability in our experiments, as we did not test on other language model families. Given the extensive nature of the experiments, which are time- and resource-intensive, we chose to focus on a specific family of models. However, we will make efforts to discuss this limitation and suggest directions for improving generalizability in the revised version.
>
> Additionally, we experimented with different coefficients when applying the steering vector to control the output, and we will include the results in the appendix of the revised version.
>
> Randomization: We agree that incorporating randomization is crucial for making our experimental results more comprehensive and reliable. We will conduct additional experiments to explore the effects of different random seeds and update the figures accordingly.
>
> We appreciate your thoughtful feedback, and we will address each of these points in our revised paper.

---

> ### Author Response · Authors · 2024-11-22
>
> Thank you for your thoughtful review. We understand the concern regarding our motivation, and we are happy to clarify why investigating when controllability emerges during pre-training is meaningful.
>
>
> Our motivation stems from the need for model training efficiency and resource optimization. As the returns from scaling model size begin to diminish, it has become crucial to explore how to effectively steer models both after training and during the pre-training phase. While previous works have mostly focused on steering fully-trained models, we believe that understanding when controllability emerges during pre-training is an important question because it can Optimize Training Efficiency, by understanding when specific concepts become steerable during pre-training, we can identify the optimal stopping points for pre-training, thus avoiding unnecessary computation and reducing resource consumption. Understanding steerability during pre-training can also help us target specific interventions that can accelerate the learning of particular concepts, making the pre-training process more effective. The emergence of steerability can serve as a tangible indicator of the model’s progress in learning specific concepts, providing a practical approach to assess the model’s development during pre-training.
>
>
> Furthermore, when training a model for specific purposes, understanding the model's ability to represent related concepts can help us determine when the language model has converged to a good distribution for those concepts. For instance, as shown in Figure 1.b, the interventions for emotions like sadness, disgust, and surprise show significant room for improvement, indicating that continued training could yield better intervention effects for these concepts. Such insights are particularly appealing for certain downstream tasks, such as a role-playing chatbot. We have added a more detailed discussion of our motivation in the introduction (lines 142 to 152).
>
>
> Regarding the terminology used for controllability, we acknowledge that our original definition may have lacked precision. We will replace the term "controllability" with "linear steerability" to clarify that we are specifically referring to the ability to steer internal concept representations through linear interventions.
>
>
> To test the generalizability of our method, we also conducted experiments on checkpoints from an additional open-source model, Amber. Due to time constraints, we were unable to conduct multiple experiments with different seeds for Amber. The ID scores for Amber can be found in Appendix A.10 (page 26).
>
>
> In our experiments, we explored different scaling factors and layers for adding the representation vector. In Appendix A.6 (pages 21 to 23), we provide a detailed exploration of different scaling factors and layer choices, which helped us identify the optimal experimental settings. For the main experiments, we used a scaling factor of 40 and applied the representation vector to the top 10 layers.
>
>
> For the supervised task, we increased the number of experimental runs and used different seeds. Figure 2 (page 3) presents the results across various datasets after applying interventions, with error bars indicating variability.

---

> > ### Comment · Reviewer_Jm63 · 2024-11-25
> >
> > Thanks for replying to my comments! I think the motivation has been strengthened, and the experimental contribution has been greatly strengthened with the random seeds, experiments on Amber, and scaling experiments. On the other hand, I've been following the other responses, and I agree with the methodological concerns of R2. I am increasing my ratings and overall score to a 5 to reflect these changes.

---

> > > ### Author Response · Authors · 2024-12-01
> > >
> > > Thank you so much for taking the time to revisit your evaluation and for your thoughtful feedback. I’m really glad to hear that the strengthened motivation and additional experimental contributions resonated with you. I also appreciate your recognition of the updates, including the random seeds, experiments on Amber, and scaling experiments. Your constructive comments and encouragement are truly motivating. I understand your agreement with the methodological concerns of R2, and I want to assure you that we are actively working to address these concerns to further strengthen our work. Thank you again for your support and for adjusting your score—it means a lot!

---

> > > > ### Author Response · Authors · 2024-12-03
> > > >
> > > > Thank you once again for your thorough review of our paper and for providing valuable suggestions. Aside from some formal errors and concerns for which results cannot be obtained in the short term, we have attempted to address most of the concerns raised by the other reviewers. We hope that the final revised paper and the ongoing discussion will more clearly convey our work and address your concerns. Thank you!

---

### Official Review · Reviewer_aCxr · 2024-11-02

**Soundness:** 2
**Presentation:** 2
**Contribution:** 3
**Rating:** 5
**Confidence:** 4

**Summary:**

The authors present the “Intervention Detector” (ID) to detect when the controllability of internal representations changes/emerges in Large Language Models (LLMs). The idea is to use a model’s hidden representations for suitable stimuli, which are inputs related to concepts such as emotions. They calculate an ID score to approximately detect when this controllability emerges during training. Empirically, they analyse the emergence of controllability across several checkpoints of the entire pre-training phase of an open-source model to pinpoint the emergence of concepts such as emotions.

**Strengths:**

The idea of detecting when high-level concepts, such as a model’s understanding of emotions, can be controlled for the first time during training (by testing when interventions show an effect) is exciting and promising.

The article is rich with graphics that present the results in an intuitive way. The latter are interesting and motivate future work.

**Weaknesses:**

Unfortunately, the article suffers from technical inaccuracies that make it very difficult to trace what exactly the authors did in their experiments.



Section 3:

Firstly, there are no definitions for $h_+, h_-$, the normalized() function, $S_{test}, A_i$, and checkpoints. The Appendix furthermore lists $H^+$ and $H^-$, which were never defined. Could the authors include a clear (sub)section for definitions, perhaps at the beginning or within an expanded notation subsection, to define all key terms? Could they also ensure consistency between the main text and the appendix regarding notation?

In step 1 of the ID method, there are positive and negative "stimuli". While the authors list a template to create positive and negative stimuli in the Appendix, it is unclear what the "{positive concept scenario}" is. Could the authors provide specific examples of positive and negative scenarios used in their experiments? Additionally, could they explain the source and selection criteria for these scenarios?

In step 2, the $v \in \mathbb{R}^{1 \times 4096}$ is supposed to be the first principal axis, correct? If so, there is no dimensionality reduction. Furthermore, the sets $S_{pos}$ and $S_{neg}$ contain positive and negative "samples", respectively. Are "samples" the stimuli from before?

At the end of step 2, the authors claim that "For a layer $l$, this vector $v_l$ is linked to a specific concept". Why? Could the authors provide empirical evidence or theoretical justification to support this claim?

In step 3, equation (3), there is a "[-1]" missing, correct?

In Step 4: Is there any theoretical justification for adding $v_l$ to a layer?



Section 4:

Firstly, what is CrystalChat? There is no reference or information. Furthermore, the authors extracted checkpoints (I guess from the pre-trained LLM360 Crystal model) and then "fine-tuned each checkpoint" - do you mean fine-tuning the model based on the weights for these checkpoints? Moreover, why fine-tune the model in the first place? It is also the first step in Figure 3, but–--from my understanding—the ID method's first step is collecting hidden states based on prompting models at different stages of the pre-training phase. Could the authors clarify this process and add a dedicated subsection within Section 4 that outlines the model architecture, training process, and rationale for fine-tuning? If the models are fine-tuned before the analysis, how can it be ensured that whatever is measured did emerge during pre-training (and not fine-tuning)?

Overall, there is no information about the experimental setup in this section. Since this paper focuses on one specific model, it would be nice to have some reference background so readers do not immediately need to consult the LLM360 paper. Furthermore, the authors use ChatGPT to evaluate the model output after intervening and list a template in the Appendix. However, it is very difficult to grasp what the included "{CrystalChat intervention results}" are. It would be helpful to include some concrete examples to showcase the contents of all the templates.

The entropy plots in Figure 5 show variations in the range 4.89 - 4.96 (or smaller). I am wondering whether these variations are significant in absolute terms. Could the authors provide some reference plots or baselines to illustrate why these variations are representative of the described behaviour?

Figure 6 shows notable differences for the checkpoints at 78.11% and sometimes at 62.81% and 93.41%. Why do these occur? What makes these checkpoints unique?

The datasets used for the Supervised Detection Task should be referenced in the text and briefly explained. Also, models are now fine-tuned with different learning rates - however, it is unclear what the other hyperparameters are (learning rate scheduler, number of epochs, batch size, etc.). In addition, how many values of runs with different seeds were averaged to gain these results?

To summarise, while the results seem interesting and the overall approach promising, the paper lacks crucial details regarding how the experiments were conducted, which prevents reproducibility. There is also no code available. However, I am open to improving my rating if the authors provide sufficient details and polish their article.




Minor:

The papers for specific Algorithms like PCA and t-SNE should be referenced; similarly, the model version (for example, for ChatGPT) should be mentioned.

Grammar and spelling mistakes need to be corrected. For example:

Line 068/069: “on model’s output” -> “[based] on a model’s output”

Line 080/081: “(Crystal (Liu et al., 2023))” -> needs rephrasing

Line 095/096: “be summarized thus:” -> “be summarized as:”

Line 145/146: “assessing model’s” -> “assessing a model’s”

Line 146/147: “figure 3” -> “Figure 3”

Line 161: “Tan et al. (2024) uses” -> “Tan et al. (2024) use”

Line 199/200: “hidden state value” -> “hidden states”

Line 210/211 and 214/215: “Detecting” -> “Detection”

Line 224/225: “at the -1 token position” -> “at the final token[’s] position”

Line 268: “Representation Vector” -> “Representation Vectors”

Line 279/280: “ID scores Across” -> “ID Scores Across”

Line 317: “Figure 4 use” -> “Figure 4 uses”

Line 318/319: “show ID score” -> “show ID scores”

Line 766/767 and 770/771: “Give the statement” -> “Given the statement” (Is this just an error in the article, or is it also present in the code for the experiments?)

**Questions:**

See Weaknesses.

---

> ### Author Response · Authors · 2024-11-13
>
> Thank you very much for your detailed and insightful review. We sincerely appreciate the time and effort you put into providing such comprehensive feedback. Your thoughtful comments reflect a deep understanding of the work and have highlighted several key areas for improvement. We are truly grateful for your constructive and responsible review, and we are committed to addressing each of your concerns.
>
> To provide an initial response, we plan to make the following changes:
>
> 1. Technical Inaccuracies: We sincerely apologize for the technical inaccuracies in Section 3. We will carefully revise this section to ensure all technical details are precise and clearly presented to address your concerns.
>
> 2. Unclear Explanations: We will add further explanations in Section 4, particularly regarding the methodology steps and the necessity of the fine-tuning process, to make the paper more accessible and easier to follow.
>
> 3. Experimental Setup: We will include a dedicated subsection detailing the experimental setup, which will improve the reproducibility of our work and ensure that our procedures are transparent.
>
> 4. Discussion on Findings: We will enhance the discussion of our findings by providing additional reference plots and a deeper analysis of the experimental results to support our conclusions more effectively.
>
> We hope that incoming revisions will address your concerns comprehensively. Once again, thank you for your valuable feedback, which has been instrumental in guiding us to improve our work.

---

> ### Author Response · Authors · 2024-11-22
>
> Thank you for your thoughtful review. We appreciate you pointing out that the definition of certain terms was not sufficiently clear. We have made a table (in Appendix B, page 27) to clarify all the notations used in the Methodology section, and we have added more context when each notation first appears.
>
> We have also provided an example of a positive/negative stimulus pair in Appendix A.1 (page 14, lines 723 to 731) to illustrate how to construct an effective stimulus pair. This example scenario was generated using ChatGPT-4. We generated 1,500 short scenarios targeting six emotions, carefully reviewed all scenarios, and manually removed irrelevant ones. Thus, the selection criteria are based on human judgment.
>
> In step 2, although we used dimensionality reduction methods like PCA, our experiments did not actually reduce the number of features. Instead, we used these methods to identify the directions of maximum variance in our stimulus pairs. To better convey this, we replaced the term "dimensionality reduction" with "linear decomposition" throughout the paper. For $S_{\text{pos}}$, $S_{\text{neg}}$, the samples represent stimuli, and$S_{\text{pos}}$, $S_{\text{neg}}$ refer to sets containing multiple stimuli.
>
> At the end of step 2, we state that "For a layer $ l $, this vector $v_l$ is linked to a specific concept." The rationale behind this statement is that when designing stimulus pairs, we aim for opposing concepts to be the main factor influencing the model's response, and we capture the direction of this concept using methods like PCA. Intuitively, the direction represented by the vector at this point corresponds to the model's understanding of the respective concept. In Appendix A.7 (page 24, lines 1250 to 1270), we demonstrate how the ratio of each PCA component changes during different pre-training stages. Towards the later stages of pre-training, the first component becomes more dominant, and this component effectively enables intervention, leading us to conclude that the vector obtained through PCA is linked to a specific concept.
>
> In step 3, we acknowledge that we missed "[-1]"—thank you for your careful attention to detail :)
>
> In step 4, while there is no theoretical guarantee regarding the effectiveness of this approach (similar to other studies on steering concepts, which also cannot provide definitive proof of intervention efficacy), we conducted multiple intervention experiments. We have added more explanatory context about the intervention process (lines 277 to 282) and provided numerous intervention results under different settings (e.g., scaling factors and layers to which the vector was added, result in Appendix A.6, page 21 to 23) to support the following points:
> 1. The vector $v_l$is linked to a specific concept.
> 2. Adding $v_l $ to a layer is an effective intervention method.
>
> For Section 4, we have added more detailed information about Crystal, CrystalChat (in Appendix A.4, page 18), and our fine-tuning settings (in Appendix A.5, page 19). "Fine-tuning each checkpoint" refers to selecting checkpoints every 15,000 steps and then fine-tuning them using the corresponding weights. Fine-tuning is crucial for our experiments because the Crystal model lacks the basic ability to follow instructions and maintain conversations. We compared the responses from the base model and the fine-tuned model at the last checkpoint (in Appendix A.5.3, page 19, lines 1015 to 1032) and found that the base model's responses were largely incomprehensible, making it difficult to confirm whether the model understood our stimuli. Although it is true that the emergence of steerability can be affected by fine-tuning, the prerequisite for our experiments was that the model should have a basic ability to understand instructions and stimuli. To minimize the impact of fine-tuning, we used only 1/10th of the data from selected datasets, all of which were dialogue datasets. We fine-tuned each checkpoint for only one epoch to provide the model with a basic understanding of instructions.
>
> We used ChatGPT-4 to evaluate the intervention results, and an evaluation template is provided at the end of Appendix A.1 (pages 14-15, lines 749 to 772).

---

> > ### Author Response · Authors · 2024-11-22
> >
> > For the entropy plots in Figure 5, the changes in entropy are not highly significant in absolute terms. To confirm the meaningfulness of the entropy change, we conducted multiple experiments and plotted the entropy with error bars. The results clearly show that a consistent decline in entropy across multiple consecutive checkpoints (i.e., higher layers showing higher ID scores, leading to entropy reduction) is followed by an increase in entropy (as more layers show higher ID scores), signifying the emergence of steerability.
> >
> > In Figure 6, we do observe notable differences for specific checkpoints. However, we would like to emphasize that this is an observed phenomenon, and we cannot definitively explain why certain checkpoints become distinct. The model's steerability for different concepts appears to be an inherent property, likely related to the pre-training dataset (i.e., the more information the dataset contains about a particular concept, the earlier the steerability for that concept emerges). Unfortunately, we have not further investigated the underlying reasons for this phenomenon.
> >
> > We have briefly introduced the datasets used in the supervised detection task (lines 481 to 483) and provided additional fine-tuning hyperparameters in Appendix A.5.2 (page 19). We ran the experiments three times with different random seeds, and the results are presented in Appendix A.8 (page 24).
> >
> > We also conducted experiments using another open-source language model, Amber, and the results are shown in Appendix A.10 (page 26).
> >
> > We hope the above responses address your concerns.

---

> > > ### Comment · Reviewer_aCxr · 2024-11-22
> > >
> > > Thank you for the detailed answer; I will read and review the revised article as soon as possible. Meanwhile, I suggest you proofread the article again to eliminate any remaining/new language errors. For example, in line 230/231:
> > >
> > > "We then pair each positive stimulus $s^+_i$ is paired with a corresponding negative stimulus $s^-_i$, forming a pair denoted as $s_i$."

---

> > > > ### Author Response · Authors · 2024-11-25
> > > >
> > > > Thank you for your valuable feedback and for pointing out the issue in lines 230/231. We appreciate your suggestion and will carefully proofread the article once again to address any remaining or newly introduced language errors. Your comments are greatly helpful, and we will ensure the revised version reflects these improvements. Please let us know if there are any additional aspects you would like us to address!

---

> > > > > ### Comment · Reviewer_aCxr · 2024-11-25
> > > > >
> > > > > **Thank you for responding to my review and the mentioned criticism. I read the article again, and I think the methodology is much clearer now than before. However, I still have concerns and questions regarding formalism, which I will enumerate below.**
> > > > >
> > > > >
> > > > > 1.You fine-tuned the checkpoints using 10% of the CrystalChat data. Was that always the same data? (Strictly speaking, you do not measure when linear steerability emerges during pre-training but when linear steerability emerges for instruction-tuned checkpoints during pre-training.)
> > > > >
> > > > > 2.The mathematical formalism still lacks. The $h_i^\pm$ in “equation” 1 should not have indices, correct? The “normalized()” map is still not explained, which I criticised before. Furthermore, it operates on the difference between hidden states for all positive ($h^+$) and negative ($h^-$) stimuli. Are these not sets? How is the difference defined in this case, when the result ($H_\mathrm{train}$) is a matrix?
> > > > >
> > > > > The ID score notation is also inconsistent. There are:
> > > > >
> > > > > $I_i^l$ (depending on a stimulus index and a layer index)
> > > > >
> > > > > $A_{i, j}$ (depending on a checkpoint index and a layer index)
> > > > >
> > > > > ID $\mathrm{score}_l$ (depending on a layer index)
> > > > >
> > > > > These are three separate but incompatible notations for the same ID score. This is confusing and prohibits reproducibility.
> > > > >
> > > > > The notation when introducing the K-Means alternative is also not well-defined. You need to be much more precise in all of these steps because your experiments are not reproducible otherwise.
> > > > >
> > > > > 3.For the unsupervised detection task, you use positive and negative stimuli that are derived from emotions. If $S_\mathrm{pos}$ consists of stimuli related to happiness as in the example in Appendix A.1, the negative concept making up $S_\mathrm{neg}$ can be any other emotion such as sadness or anger (line 720/721). Does that mean that $S_\mathrm{neg}$ contains a balanced mix of stimuli from all emotions but happiness? Furthermore, you write in line 732/733 that “For each experiment, we randomly select 256 stimulus pairs and divide them into a training dataset $S_\mathrm{train}$ and a test dataset $S_\mathrm{test}$.” However, $H_\mathrm{train}$ is an element in $\mathbb{R}^{256 \times 4096}$ - is this the same “256”, meaning $H_\mathrm{train}$ contains the information of training and test stimuli?
> > > > >
> > > > > 4.According to equation 3, the ID score is determined by taking the Euclidean inner product of the final hidden state of the model and the representation vector produced by either PCA or K-Means, correct? Why did you not normalise both vectors so that the ID score effectively becomes the cosine similarity? The problem I see with this definition is that the length of the vectors has an influence on the results (a bias). Can you explain or mathematically justify this design decision? The reason I mention this point is that you write in line 366/367: “For example, before 48% of pre-training, anger representations are mostly noise across layers.” However, this is not necessarily true, given the definition in equation 3. The values are lower, yes, but this does not mean that the representations are noisy. In contrast, if you had used the cosine similarity of the vectors, you would have a well-defined base value, namely 0, which indicates no alignment of representations (as the vectors would be orthogonal). Without the normalisation, you can still detect trends (as indicated by the heatmaps) but not deduce that representations reduce to noise.
> > > > >
> > > > >
> > > > >
> > > > > 5.You tested the scaling factors for the intervention mentioned in Step 4. with the fully instruction-tuned CrystalChat, but used them for the models that were only instruction-tuned with 10% of the data, correct?
> > > > >
> > > > >
> > > > > 6.I am also not convinced by the statement in line 312/313: “However, once multiple layers achieve strong alignment, the decline in entropy slows down or even reverses.” If the decline in entropy reverses, entropy increases, correct? How does that conceptually fit into layers being strongly aligned? Intuitively, the stronger the alignment, the less entropy, as alignment allows inferring properties of a second layer from a first. Can you explain this sentence/hypothesis in more detail?
> > > > >
> > > > > 7.The results in Fig. 5 still do not convince me. I acknowledge that it is difficult to obtain a large sample size for these experiments, but the error bars seem too large to allow a conclusion. As I mentioned in my review: “I am wondering whether these variations are significant in absolute terms. Could the authors provide some reference plots or baselines to illustrate why these variations are representative of the described behaviour?”

---

> > > > > > ### Comment · Reviewer_aCxr · 2024-11-25
> > > > > > **-continued-**
> > > > > >
> > > > > > 8.While the results in Fig. 7 seem interesting on a surface level, they could also just show that representations for any concept are different for earlier checkpoints than for later checkpoints. This means that you could receive the same results for concepts that are not emotions or are not meaningful at all. Similar to the previous point, I suggest you add plots that result from the same methodology but for entirely different concepts as baselines.
> > > > > >
> > > > > > 9.Finally, the ID score can be computed using PCA or K-Means - but what did you use in your experiments?
> > > > > >
> > > > > >
> > > > > > In conclusion, I cannot change my rating yet since too many issues still need to be addressed. I acknowledge and appreciate the work that went into the article and, especially, its revision, which did improve the clarity. (There are still numerous spelling mistakes in the article, and some of the newly added results need better descriptions, for example, in Table 3: what does the “#” indicate?).
> > > > > >
> > > > > > Nevertheless, I think this work is interesting and the approach unique, so I want to motivate the authors to re-submit to a prestigious conference/journal once the issues have been addressed.

---

> > > > > > > ### Author Response · Authors · 2024-12-01
> > > > > > >
> > > > > > > We sincerely thank you for taking the time to carefully review our paper again and for providing such specific and detailed concerns. Although we were unable to upload a revised PDF version within the review timeline, we deeply appreciate your thoughtful feedback and effort. Even though the score was not adjusted, we value your insights and are grateful for the opportunity to address your concerns here.
> > > > > > >
> > > > > > > Below, we provide detailed explanations to clarify and respond to the concerns you raised:
> > > > > > >
> > > > > > > 1. We used exactly the same dataset to fine-tune each checkpoint. It is reasonable to raise the concern that our evaluation of linear steerability is conducted on instruction-fine-tuned checkpoints rather than purely pre-trained models. We acknowledge that the fine-tuning process can indeed enhance the model’s ability to understand specific concepts. Technically speaking, the emergence of linear steerability is the result of a combination of both fine-tuning and pre-training. However, this fine-tuning process is essential because there is no definitive "correct" answer for emotions. Unlike supervised tasks, where ground truth answers enable evaluation through logits or other metrics, emotions cannot be evaluated in the same way. Instead, we can only confirm that a model understands a concept by assessing its responses. Therefore, enabling the model to follow instructions and provide interpretable, readable responses is critical for our experiments. We also tested how much the fine-tuning process contributes to the model’s performance on supervised tasks. The results indicate that while fine-tuning does improve accuracy, the improvement is very limited. This suggests that the primary effect of fine-tuning lies in enabling the model to follow instructions rather than significantly enhancing its conceptual understanding.
> > > > > > >
> > > > > > > | Training Percentage | Base Model (%) | Fine-tuned Model (%) |
> > > > > > > |-|-|-|
> > > > > > > | 7.00%| 37.8%| 40.5% |
> > > > > > > | 13.99%| 39.2% | 41.4%|
> > > > > > > | 20.99% | 40.9% | 42.2% |
> > > > > > > | 27.99% | 41.1%  | 41.1% |
> > > > > > > | 34.98% | 41.5% | 41.9%
> > > > > > > | 41.98% | 42.4%| 43.4%|
> > > > > > > | 50.38% | 41.6%| 43%|
> > > > > > > | 55.97% | 43.2%| 44.3%|
> > > > > > > | 62.97% | 42.1%  | 43.%|
> > > > > > > | 69.97% | 41.7% | 43.9%|
> > > > > > > | 78.36% | 45.8%| 46.8%|
> > > > > > > | 83.96% | 45.2% | 47.4% |
> > > > > > > | 90.96%| 46.4% | 47.7% |
> > > > > > > | 95.16%| 48% | 49.9%  |
> > > > > > > | 100.00% | 47.3% | 49.4% |
> > > > > > >
> > > > > > > 2. We would like to clarify that the $i$ in $h_i^{\pm}$ is already an index in our notation. Specifically, $i$ represents the $i$-th stimulus pair ($s_i^+, s_i^-$) in the dataset $S$. To elaborate, the hidden states $h_i^+$ and $h_i^-$ are defined as: $h_i^+ = R(M, s_i^+)[-1], \quad h_i^- = R(M, s_i^-)[-1], $where $R(M, s)$ denotes the hidden states output by the model $M$ when the stimulus $s$ is input, and $[-1]$ refers to the final token position. Including $i$ as an index is essential for distinguishing between different stimulus pairs in the dataset. It ensures that the normalization and subsequent steps, such as PCA, are applied independently to each pair. Without this indexing, it would be unclear that these operations are performed on a per-sample basis.
> > > > > > >
> > > > > > > In our methodology, the $ \texttt{normalized()} $ function refers to a Z-Score normalization process
> > > > > > > applied independently to each layer of the model. Specifically, for a given layer's hidden state differences ($ h^+ - h^- $), we compute the mean ($ \mu $) and standard deviation ($ \sigma $) of the values within that layer, and normalize them using the following formula:
> > > > > > > $ \text{normalized}(x) = \frac{x - \mu}{\sigma}. $
> > > > > > > This ensures that the data within each layer has a mean of 0 and a standard deviation of 1. The primary purpose of this normalization is to eliminate the influence of scale disparities between different features or layers, which is critical for downstream linear decomposition techniques like PCA. $h^+$ and $h^-$ are individual hidden state vectors corresponding to the positive ($s_i^+$) and negative ($s_i^-$) stimuli of the $i$-th stimulus pair from the dataset $S$. Specifically, $i$ indicates the index of a stimulus pair in $S$. The hidden states are computed as follows: $h_i^+ = R(M, s_i^+)[-1], \quad h_i^- = R(M, s_i^-)[-1],$ where $R(M, s)$ denotes the hidden states output by the model $M$ when the stimulus $s$ is input, and $[-1]$ refers to the final token position. The difference $h^+ - h^-$ is calculated element-wise between these vectors for each stimulus pair, resulting in a difference vector. When extended to the entire dataset $S$, the result is a matrix $H_{\text{train}}$, where each row corresponds to the difference vector for a single stimulus pair. Mathematically: $H_{\text{train}} = \begin{bmatrix} (h_1^+ - h_1^-) \\ (h_2^+ - h_2^-) \\ \vdots \\ (h_n^+ - h_n^-) \end{bmatrix}, $where $n$ is the number of stimulus pairs in $S$. Thus, $h^+$ and $h^-$ are not sets, but rather single hidden state vectors. The term $H_{\text{train}}$ is a collection of these pairwise differences, organized as a matrix for subsequent normalization and analysis.

---

> > > > > > > > ### Author Response · Authors · 2024-12-01
> > > > > > > >
> > > > > > > > We sincerely apologize for the inconsistency in the definitions of some indices due to the excessive use of indices (layer, checkpoint, stimulus) across different methods. For example, in the PCA method,
> > > > > > > > $𝑖$ represents stimuli, while in the K-Means method, $i$ represents checkpoints. In future revisions, we will use $𝑐$ to represent checkpoints to ensure consistency. We would like to emphasize that our experiments are not reproducible. On the contrary, we have also conducted experiments on the Amber model and observed similar results, even when the stimuli sets correspond to entirely different emotional scenarios.
> > > > > > > >
> > > > > > > >
> > > > > > > > 3.$S_{\text{neg}}$ is indeed constructed as a balanced mix of stimuli representing all emotions except for happiness, ensuring diversity and balance in the negative concept. $H_{\text{train}}$ only contains the information of the training stimuli. The test set and training set are composed of completely distinct sets of 256 randomly selected stimuli pairs. There is no overlap between the training set ($S_{\text{train}}$) and the test set ($S_{\text{test}}$). This ensures the independence of evaluation and avoids information leakage between training and testing.
> > > > > > > >
> > > > > > > >
> > > > > > > >
> > > > > > > > 4.We chose this design intentionally because the magnitude of the vector contains important information about the alignment and activation strength in relation to the extracted concept direction.
> > > > > > > >
> > > > > > > > Concepts in the activation layers are inherently directional, and PCA or K-Means is used to identify these concept directions. The inner product is then used to compute the projection of each layer's hidden state onto the concept direction. This projection reflects not only the alignment (direction similarity) but also the activation strength (magnitude) in the concept direction. A higher ID score indicates a stronger projection, which implies both better alignment and stronger activation for the concept.
> > > > > > > >
> > > > > > > > If cosine similarity were used instead, the magnitude information would be lost, and the ID score would only reflect direction similarity. This would prevent us from capturing the full extent of each layer's representation of the concept, as magnitude plays a crucial role in measuring the strength of the alignment.
> > > > > > > >
> > > > > > > >  A low ID score does not directly indicate that the representations are "noise." Instead, it reflects that the representations are not well-aligned with the correct concept direction.
> > > > > > > >
> > > > > > > > In our methodology, the concept direction is determined through PCA or K-Means, which aims to capture meaningful patterns within the hidden states. If a layer's representation fails to align with this direction (as indicated by a low ID score), it suggests that the layer's activations do not strongly project onto the identified concept.
> > > > > > > >
> > > > > > > > Furthermore, if we perform interventions using a "concept" that is not well-aligned (i.e., derived from a low ID score), the resulting vector lacks clear interpretability in human-understandable terms. In this sense, such misaligned directions can be considered noise, as we cannot definitively relate them to any specific concept. This reasoning underpins our interpretation of low ID scores and their connection to noisy or unaligned representations.
> > > > > > > >
> > > > > > > > 5.Yes, that is correct. We tested the scaling factors for the intervention mentioned in Step 4 with the fully instruction-tuned CrystalChat model and then applied the same scaling factors to models that were only instruction-tuned with 10% of the data.
> > > > > > > >
> > > > > > > > 6.In the early stage of pretraining, the ID scores are generally low and distributed relatively evenly across layers, resulting in high entropy. As training progresses, some layers start to achieve higher ID scores due to improved alignment with the correct concept direction. This leads to a more concentrated distribution of ID scores, reducing entropy.
> > > > > > > >
> > > > > > > > However, in the later stage of pretraining, as more and more layers achieve high ID scores, the distribution becomes uniformly high. This uniformity can cause entropy to increase again, even though the ID scores themselves indicate strong alignment across layers. This dynamic reflects the transition from a random distribution (early stage) to a concentrated distribution (mid stage), and finally to a uniform high-value distribution (late stage).

---

> > > > > > > > ### Author Response · Authors · 2024-12-01
> > > > > > > >
> > > > > > > > 7.Thank you for your valuable feedback regarding the results in Figure 5. We agree that the error bars are indeed relatively large, which could be attributed to the limited sample size in these experiments. This is an area we are actively working to improve.
> > > > > > > >
> > > > > > > > We also appreciate your suggestion to provide reference plots or baselines to better contextualize these variations. However, at this stage, we have not yet identified a suitable method to construct meaningful reference plots or baselines for this specific task. We acknowledge the importance of this and will explore potential approaches in future work to strengthen the interpretability and reliability of our results.
> > > > > > > >
> > > > > > > > 8. We totally agree that the representation differences between earlier and later checkpoints are likely a general phenomenon and not unique to emotional concepts. This distinction may also exist for other concepts.
> > > > > > > >
> > > > > > > > The novelty of our work lies in observing that even for seemingly related concepts, such as "happy" and "sad," their emerge times can be entirely different. Based on this observation, we attempt to identify indicators that can explain this phenomenon or even predict the time at which steerability emerges during pretraining.
> > > > > > > >
> > > > > > > > We acknowledge that emotional concepts are not inherently unique, and similar trends in representation dynamics could be observed for other types of concepts. However, as shown in Figure 16 (Page 28), we find an intriguing pattern: concepts with earlier-emerging steerability tend to exhibit better separation between representations in late and early checkpoints, whereas concepts with later-emerging steerability show the opposite trend. This horizontal comparison across emotion concepts provides meaningful insights into their dynamics during pretraining.
> > > > > > > >
> > > > > > > > 9. In all the experiments shown in the figures, we used PCA because it provides results that are both intuitive and easy to interpret. While K-Means can also generate similar ID score heatmap plots, the results are less visually clear due to the increased noise associated with this method.
> > > > > > > >
> > > > > > > > We sincerely appreciate your recognition of the novelty and interesting aspects of this work, as well as your effort in revisiting our paper and actively engaging in the discussion. Although the score was not adjusted, your attention to detail has helped us identify areas where the paper can be further polished. We will continue to work on this paper and strive to make the results even more robust and comprehensive. Thank you again :)

---

> > > > > > > > > ### Comment · Reviewer_aCxr · 2024-12-02
> > > > > > > > >
> > > > > > > > > Thank you again for the thoughtful response. I appreciate you addressing my points of critique and answering my questions (I did not change my score earlier because I waited for this response). The details you provided make the results much more credible, though I'm afraid I still have to disagree with some of the mathematical design choices, such as using the inner product instead of the normalised cosine similarity for the ID score. However, this article has undergone a significant transformation from its submitted version and has greatly improved in quality. This I will honour by improving my rating, although I cannot yet agree with accepting it. Nevertheless, I am certain that the next version will be accepted once the remaining changes have been made.
> > > > > > > > >
> > > > > > > > > P.S. You write in your response to point 2: "*We would like to emphasize that our experiments are not reproducible.*" I take it this was an error.

---

### Official Review · Reviewer_ecWs · 2024-11-05

**Soundness:** 3
**Presentation:** 3
**Contribution:** 3
**Rating:** 6
**Confidence:** 4

**Summary:**

The current paper investigates the development of controllability in language models's pre-training. It focuses on a very specific type of control, i.e., intervention, and the control of a very specific kind of factors, i.e., concepts related to emotions. The authors introduce the "Intervention Detector" (ID) as a method to track how control over specific concepts emerges and solidifies during pre-training. By using dimensionality reduction techniques on hidden states, ID identifies points at which different concepts—such as emotions and reasoning—become extractable and controllable.

The study reveals that concepts don’t become controllable simultaneously; for example, anger is controllable earlier in pre-training than sadness. The authors validate these findings with metrics like ID scores and heatmap visualizations, showing that as pre-training progresses, the model's hidden states align more with the extracted concepts, allowing for more effective intervention.

**Strengths:**

- I find this paper is working on a very interesting topic that is worth investigating, i.e., when the controllability emerges during pre-training. It is not only valuable to people working on tasks like knowledge editing, but also helps us understand the learning procedures of LLMs.
- One interesting finding that I like is how the control of different (emotional) concepts emerge differently from each other. Maybe it is worth check whether this is consistent with human beings.

**Weaknesses:**

As a paper that investigates the "controllability" (as its title suggests), though the paper has many interesting findings, I expected it to consider more control techniques and factors, whereas the current paper focuses on a very specific type of control, i.e., intervention, and the control of a very specific kind of factors, i.e., concepts related to emotions.

Another major risk of the paper is its writing, making the paper hard to follow. I am saying this with the following concerns:
- The primary ability of LLMs is to generate language, and, in this context, the term "controllability" could have multiple meanings for me and it could be one in many ways, prompting, instruction tuning, etc. It appears that what this paper focuses on is actually the "editability" (instead of the wider controllability) of LLMs. I suggest the authors first define what "controllability" means in this paper.
- The above point makes many terms, especially in the abstract and introduction, hard to interpret. For example, it is hard to understand how a concept can be controlled.
- More explanations are needed for the causality between the scores you computed and the controllability, e.g... what is SNR used for?

**Questions:**

See my comments in "weaknesses".

---

> ### Author Response · Authors · 2024-11-13
>
> Thank you very much for your thoughtful review and valuable feedback.
>
> We are committed to addressing your concerns with the following actions:
>
> 1. Clarify the Scope of Controllability: We sincerely appreciate your observation regarding the term "controllability." We understand that it may lead to multiple interpretations and potential misunderstandings. We agree that the term is not entirely accurate for describing our findings and will consider using more precise alternatives, such as "editability" or "linear steerability," in the revised version.
>
> 2. Improve Clarity: We acknowledge the need for clearer explanations, particularly concerning the ID scores and other key terms in the paper. We will provide additional explanations to ensure our methodology and findings are more accessible and easier to follow.
>
> We are currently working on these revisions and will upload an updated version that fully addresses your concerns. Your feedback has been instrumental in helping us improve the quality of our work, and we thank you again for your time and constructive suggestions.

---

> ### Author Response · Authors · 2024-11-22
>
> Thank you for pointing out the imprecision in our use of the term "controllability." We appreciate your feedback and have made adjustments accordingly.
>
>
> Regarding the terminology, we will replace the term "controllability" with "linear steerability" to clarify that we are specifically referring to the ability to steer internal concept representations through linear interventions. In both the introduction and abstract, we have revised the terminology to reflect this more accurately, emphasizing that our method focuses on linear steerability.
> To give a simpler explanation of our method, we use linear algebraic techniques, such as PCA, to identify a direction in the hidden state space that corresponds to a particular concept, such as an emotion. By adding this vector during the inference process, we can effectively steer the model's output to either enhance or diminish the presence of that concept. In this manner, we influence the internal representations to achieve the desired expression in the generated language, which is why we use the term "linear steerability" rather than "control." This steerability allows us to subtly guide the model's behavior to better align with specific concepts we intend it to represent.
>
>
> We have explained the relationship between ID score and effective interventions in lines 283 to 285, as well as the rationale for using the SNR (Signal-to-Noise Ratio) in lines 286 to 293. A higher ID score can be interpreted as indicating a higher SNR for the specific concept represented by the extracted vector. Therefore, a higher ID score suggests that the concept represented by the vector aligns more closely with the intended human-understandable concept, rather than being random noise. In Appendix A.7 (page 24), we present the ratios of the first five components obtained through PCA across multiple checkpoints. It can be observed that the ratio of the first component increases progressively during pre-training. Since we interpret the direction of the first PCA component as representing the concept direction, an increase in the signal component (i.e., the effective part of the vector) and a decrease in the noise component indicate a more effective intervention.

---

> > ### Comment · Reviewer_ecWs · 2024-11-26
> >
> > Thanks for your hard work. I found the revision to be much clearer. I raised my score for the presentation but kept my overall score (because I partly agree with the concerns of the other reviewers and because there is no 7, which is very weird).

---

> > > ### Author Response · Authors · 2024-12-01
> > >
> > > Thank you so much for your feedback. I’m glad to know that the revisions made the content clearer and that you found the updates helpful. I completely understand your decision not to raise the overall score, and I genuinely appreciate your careful consideration and balanced evaluation. Your encouragement and constructive comments are truly valuable to me. Thank you again!

---

> > > ### Author Response · Authors · 2024-12-03
> > >
> > > Thank you once again for your thorough review of our paper and for providing valuable suggestions. Aside from some formal errors and concerns for which results cannot be obtained in the short term, we have attempted to address most of the concerns raised by the other reviewers. We hope that the final revised paper and the ongoing discussion will more clearly convey our work and address your concerns. Thank you!

---

### Author Response · Authors · 2024-11-13
**Sincere Acknowledgment of Reviewer Comments**

We sincerely appreciate the time and effort each of you has taken to review our paper. Your constructive comments and valuable suggestions have provided us with meaningful insights and directions for improving our work.

We are currently working on a thorough revision to address the concerns and questions raised in your reviews. We are carefully considering each point you highlighted and making the necessary adjustments to enhance the clarity, rigor, and overall quality of our paper.

Thank you again for your thoughtful feedback. We are committed to ensuring that our revisions meet your expectations and strengthen the contribution of our research：）

---

### Author Response · Authors · 2024-11-22
**Clarifications and Revisions on the Use of ‘Controllability’ and ‘Dimensionality Reduction’ in the paper**

In our paper, we revised the terminology for two key concepts. Compared to the original term "controllability," the term "linear steerability" more accurately reflects our methodology, as our approach to control focuses on utilizing linear methods to steer the internal representations of concepts in the model. Secondly, we updated the description of "dimensionality reduction." While we employed PCA, we did not reduce the number of features. Instead, we used PCA as a dimensionality reduction technique to identify the direction of maximum variance within the stimulus set. To better capture this difference, we replaced "dimensionality reduction" with "linear decomposition." We believe these more precise terms enhance the clarity and interpretability of the paper.

---

### Meta-Review · Program_Chairs · 2024-12-24

**Metareview:**

PC is entering meta-review on behalf of SAC/AC:

 The paper had middling scores, and reviewers felt that the paper both lacked clarity on practical details, and had either inaccuracies or vagueness that made it unsuitable for publication in its current form.

**Additional Comments On Reviewer Discussion:**

TBD

---

### Decision · Program_Chairs · 2025-01-22

Reject